EMBO
Molecular Medicine

# Mouse *Nr2f1* haploinsufficiency unveils new pathological mechanisms of a human optic atrophy syndrome

Michele Bertacchi[1,2,*] , Agnès Gruart[3], Polynikis Kaimakis[4,5], Cécile Allet[6,7], Linda Serra[1,8], Paolo Giacobini[6,7] , José M Delgado-García[3], Paola Bovolenta[4,5] & Michèle Studer[1,**]

## Abstract

Optic nerve atrophy represents the most common form of hereditary optic neuropathies leading to vision impairment. The recently described Bosch-Boonstra-Schaaf optic atrophy (BBSOA) syndrome denotes an autosomal dominant genetic form of neuropathy caused by mutations or deletions in the *NR2F1* gene. Herein, we describe a mouse model recapitulating key features of BBSOA patients—optic nerve atrophy, optic disc anomalies, and visual deficits—thus representing the only available mouse model for this syndrome. Notably, *Nr2f1*-deficient optic nerves develop an imbalance between oligodendrocytes and astrocytes leading to postnatal hypomyelination and astrogliosis. Adult heterozygous mice display a slower optic axonal conduction velocity from the retina to high-order visual centers together with associative visual learning deficits. Importantly, some of these clinical features, such the optic nerve hypomyelination, could be rescued by chemical drug treatment in early postnatal life. Overall, our data shed new insights into the cellular mechanisms of optic nerve atrophy in BBSOA patients and open a promising avenue for future therapeutic approaches.

**Keywords** astrogliosis; BBSOA syndrome; mouse *Nr2f1*; myelination; optic nerve atrophy

**Subject Categories** Development & Differentiation

## Introduction

Optic atrophy denotes the loss of part or all the nerve fibers in the optic nerve (ON), often leading to widening of the optic cup, and represents an important sign of advanced ON disease frequently associated with gradual vision loss or reduced visual acuity. Optic neuropathies may range from non-syndromic genetic diseases to rare syndromic multisystemic disorders. The most common forms of inherited optic neuropathies, described so far, are the Leber's optic neuropathy (LHON), and the dominant optic atrophy (DOA) caused by mutations in the nuclear gene OPA1 (Carelli *et al*, 2017; Chun & Rizzo, 2017). Recently, patients carrying deletions or missense point mutations in the *NR2F1* locus have also been diagnosed with optic atrophy associated with developmental delay and intellectual disability (Al-Kateb *et al*, 2013; Bosch *et al*, 2014; Chen *et al*, 2016; Kaiwar *et al*, 2017). This autosomal dominant disorder resulting from *NR2F1* haploinsufficiency is currently named as Bosch-Boonstra-Schaaf optic atrophy (BBSOA) syndrome (OMIM: 615722). BBSOA patients display a variable array of clinical deficits, both visual and cognitive, where malformed optic disc (OD), ON atrophy, decreased visual acuity, developmental delay, epilepsy, and mild-to-moderate intellectual disability are among the most common deficiencies (reviewed in Bertacchi *et al*, 2018). The clinical features of BBSOA syndrome are still evolving, as the number of reported cases is continuously increasing since 2014, when the first patients with missense mutations were reported (Bosch *et al*, 2014; Chen *et al*, 2016; Kaiwar *et al*, 2017; Martin-Hernandez *et al*, 2018). This suggests that the prevalence of BBSOA syndrome might be still underestimated, which prompted us to further understand the mechanisms of this newly identified neurodevelopmental disease.

NR2F1, also known as COUP-TFI, is an orphan nuclear receptor belonging to the superfamily of steroid/thyroid hormone receptors

1 CNRS, Inserm, iBV, Université Côte d'Azur, Nice, France
2 Fondazione IRCCS Istituto Neurologico Carlo Besta, Milan, Italy
3 Division of Neurosciences, Pablo de Olavide University, Seville, Spain
4 Centro de Biología Molecular "Severo Ochoa", CSIC-UAM, Madrid, Spain
5 CIBER de Enfermedades Raras (CIBERER), Campus de la Universidad Autónoma de Madrid, Madrid, Spain
6 Jean-Pierre Aubert Research Center (JPArc), Laboratory of Development and Plasticity of the Neuroendocrine Brain, UMR-S 1172, Inserm, Lille, France
7 University of Lille, FHU 1,000 Days for Health, Lille, France
8 Department of Biotechnology and Biological Sciences, University of Milano-Bicocca, Milano, Italy
*Corresponding author. Tel: +33 489150723; E-mail: Michele.BERTACCHI@univ-cotedazur.fr
**Corresponding author. Tel: +33 489150720; E-mail: Michele.STUDER@univ-cotedazur.fr

and acting as a strong transcriptional regulator (Alfano et al, 2013; Bertacchi et al, 2018). Two major homologs of this family have been identified in vertebrates: NR2F1 and NR2F2 (also named COUP-TFII; Ritchie et al, 1990; Wang et al, 1991; Alfano et al, 2013). Their molecular structure encompasses two highly conserved domains, the DNA-binding domain (DBD) and the ligand-binding domain (LBD). Most of the pathogenic mutations causing BBSOA syndrome are located in the NR2F1 DBD, hence disrupting its activity as a transcription factor, but some have been reported also in the LBD or in the start codon (ATG; Bosch et al, 2014; Chen et al, 2016; Bertacchi et al, 2018). The mouse Nr2f1 and human NR2F1 proteins are very well conserved during evolution and share 98–100% of sequence homology in both DBDs and LBDs (Bertacchi et al, 2018). Nr2f1 is widely and dynamically expressed in several mouse brain regions (Wang et al, 1991; Qiu et al, 1995; Tripodi et al, 2004; Armentano et al, 2006; Lodato et al, 2011b; Alfano et al, 2013; Flore et al, 2017; Parisot et al, 2017; Bertacchi et al, 2018), and its expression pattern seems to be well conserved in human embryos and fetuses, as recently shown (Alzu'bi et al, 2017a,b). Thus, structural and expression similarities between mouse and humans strongly suggest a conserved role of NR2F1 during development of the central nervous system (CNS).

Multiple data in mice have highlighted the multi-faceted functions of Nr2f1 in the development of several mouse brain structures (Alfano et al, 2013; Bertacchi et al, 2018), but its exact role during eye development is still vague. Indeed, there is still inconsistency between the ocular phenotype obtained in mouse and the clinical features described in BSSOA patients, as the ON atrophy and cerebral visual deficits identified in several patients have not been reproduced in mice lacking solely Nr2f1 (Tang et al, 2015; Bertacchi et al, 2018). Only the combined inactivation of both homologs, Nr2f1 and Nr2f2, produced severe early ocular defects, such as coloboma and microphthalmia, suggesting a genetic compensation in the mouse (Tang et al, 2010, 2015). This is surprising since BBSOA patients are haploinsufficient for NR2F1, i.e., lack only one copy of the gene, but develop ocular impairments with high prevalence (Chen et al, 2016; Bertacchi et al, 2018). This discrepancy could depend either on species-specific functional differences or on the conditional mouse model used by Tang and colleagues (Swindell et al, 2006; Tang et al, 2010). Since genetically modified mice still offer a unique opportunity to decipher the mechanisms underlying eye development and assembly of the visual pathway, we investigated the role of Nr2f1 during visual development from the retina to the visual cortex, using heterozygotes (HET) and homozygotes (KO or null) of the constitutive mouse knock-out (KO) model (Armentano et al, 2006). We reasoned that constitutive loss of one Nr2f1 allele would better reproduce the human disease condition, in which Nr2f1 dosage is decreased in all cells and from the earliest stages of development.

In this study, we report that Nr2f1/NR2F1 is expressed in the peripheral visual system in both mice and humans, particularly in cell types involved in the development and maturation of the ON, such as neural retina cells, ON astrocytes, and oligodendrocytes. Furthermore, we show that Nr2f1 HET and KO mice have clear ocular abnormalities, from OD malformations, delayed retinal ganglion cell (RGC) differentiation and apoptosis to decreased ON myelination and increased astrogliosis, resulting in reduced axonal conduction velocity from the retina to higher order centers. At adult stages, Nr2f1 HET mice have visual and associative learning deficits, reproducing in some ways the cerebral visual impairment described in patients (Bosch et al, 2014; Chen et al, 2016). Notably, Miconazole treatment in early postnatal pups rescues the ON demyelination defect, by restoring appropriate levels of oligodendrocytes, but has little effect on astrogliosis, indicating that the two events are independently controlled in this optic atrophy syndrome. Overall, we show that Nr2f1 mutant mice can be used as a model to reproduce the BBSOA syndrome and, more broadly, could serve as a tool to test possible therapeutic approaches aimed at counteracting ON neuropathies.

## Results

### NR2F1 is dynamically expressed in the mouse and human neural retina (NR) and optic nerve

As most BBSOA patients develop ON atrophy, we carefully assessed NR2F1 expression in both the mouse and human developing eye and ON. As previously reported (Tang et al, 2010), the murine Nr2f1 protein is strongly expressed in both the presumptive dorsal and ventral optic stalk (OS), the precursor of the ON at embryonic day (E) 10.5, and shows a ventral-high to dorsal-low gradient in NR progenitors (Fig 1A). At E12.5, when the optic vesicle invaginates forming a bi-layered optic cup, Nr2f1 graded expression is maintained in the ventral retina, in the OS, and in the optic disc (OD; Fig 1B–B"). Nr2f1 co-localizes with virtually all Sox2$^+$ retinal progenitors at E13.5 (Fig 1C–C") and with Brn3a$^+$ early differentiating RGCs (Fig 1D–D"), even if at lower levels when compared to progenitors (Fig 1C'–D"). At E18.5 and postnatal (P) stages, Nr2f1 is still expressed in both Brn3a$^+$ RGC neurons forming long-distance projections constituting the ON (Fig 1E–F"), and in residual progenitors (Sox2$^+$ cells; Fig 1G and G'). Beside retinal neurons, Nr2f1 is also localized in astrocytes and oligodendrocytes of the ON. Between E13.5 and E18.5, Nr2f1 is expressed in virtually all Glast$^+$/NF1A$^+$ astrocytes of the OS/ON (Figs 1H–I" and EV1A–A"), then maintained in 70% of Glast$^+$/NF1A$^+$ astrocytes at P7 and P28 (Fig EV1B), and in around 80% of Sox10$^+$ oligodendrocytes at postnatal stages (Figs 1J–K" and EV1C). In summary, Nr2f1 is highly expressed in both retinal and optic nerve components of the peripheral visual system in mouse.

Staining of NR2F1 on human embryonic sections of gestational week (GW) 11 confirmed expression conservation in both the developing neural retina (identified by the retinal marker PAX6) and the ON (Fig 1L–L"). Notably, NR2F1 is highly expressed in progenitors (SOX2$^+$) as well as in post-mitotic differentiated retinal cells (SOX2$^-$; Fig EV1D–D"). Along the ON, from the optic disc (distal) to the chiasm (proximal) regions, almost all cells are positive for NR2F1 and 93% of them co-express the astrocytic marker S100β (Fig 1M–N"). High co-expression of NR2F1 with SOX2 and S100β (Figs 1O–P' and EV1E–E"), as well as with BRN3a in the ganglion cell layer (GCL) of the neural retina (Fig 1Q–Q"'), is maintained at GW14, indicating that NR2F1 expression follows cell differentiation in both the NR (from progenitor to post-mitotic RGCs) and the ON. Moreover, oligodendrocytes have not reached the ON at these early stages, but can be distinguished in the preoptic area, where they are generated and co-express NR2F1 (93% double-positive cells;

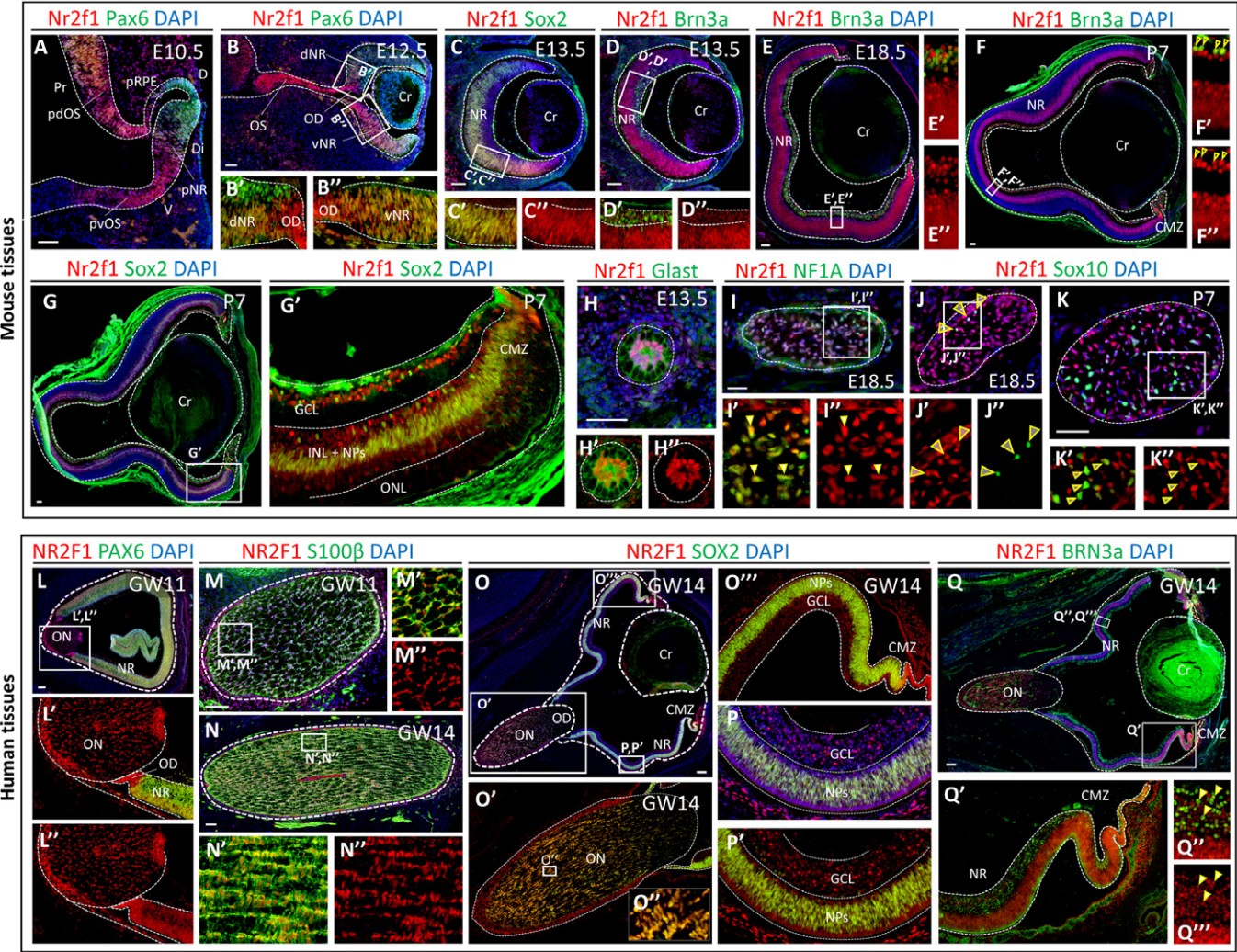

**Figure 1. Nr2f1/NR2F1 expression during neural retina and optic nerve development in mice and humans.**

A–B" Nr2f1 (red) and Pax6 (green, retinal progenitors) immunofluorescences (IF) on sagittal sections of E10.5 and E12.5 mouse optic vesicles. Note the high-ventral to low-dorsal Nr2f1 gradient in the presumptive neural retina (pNR), and high Nr2f1 expression in both the presumptive dorsal and ventral optic stalk (pvOS and pdOS). Cr, crystal lens; Di, distal; Pr, proximal; presumptive retinal pigmented epithelium (pRPE).

C–D" IF on E13.5 mouse eye sagittal sections with Nr2f1 (red) and Sox2 (green, progenitors in C, C') or Brn3a (green, RGCs in D, D') showing Nr2f1 expression in NR progenitors and post-mitotic RGCs (insets in C'–D").

E–F" Nr2f1 (red) and Brn3a (green) IF on E18.5 and P7 mouse eyes depicting Nr2f1 expression in virtually all RGCs in the ganglion cell layer (GCL). Arrowheads point to double-labeled cells.

G, G' Nr2f1 (red) and Sox2 (green) IF on P7 mouse retina illustrating low Nr2f1 expression in the inner nuclear layer (INL) and progenitors (NPs), and high levels in the GCL and ciliary marginal zone (CMZ). No expression in the outer nuclear layer (ONL).

H–K" IF on cross-sections of E13.5 optic stalks (OS), and E18.5 and P7 optic nerves (ONs) with Nr2f1 (red) and Glast (green, astrocytic progenitors in H–H"), NF1A (green, astrocytes in I–I"), or Sox10 (green, oligodendrocyte precursors in J–K") showing Nr2f1 expression in both astrocytic and oligodendrocytic lineages. Arrowheads point to double-labeled cells. See Fig EV1B and C for quantification.

L–L" NR2F1 (red) and PAX6 (green; NR domain) IF on sagittal sections of human eye primordia at gestational week (GW) 11 illustrating high NR2F1 expression in both ON and NR cells.

M–N" NR2F1 (red) and S100β (green, astrocytes) IF on cross-sections of human GW11 and GW14 ONs. Higher magnifications (M'–N") show that most of S100β+ astrocytes co-express NR2F1. Lower magnification views are shown in Fig EV1E–E".

O–Q'" NR2F1 (red) and SOX2 (green in O–P') or BRN3a (green in Q–Q'") IF on sagittal sections of human GW14 eyes indicating high NR2F1 expression in virtually all NR progenitors (O'", P, P'), differentiating RGCs (Q–Q'"), and in the majority of ON astrocytic progenitors (O'). Arrowheads point to double-labeled cells.

Data information: Nuclei counterstaining (blue) was obtained with DAPI. Scale bars: 50 or 100 µm for mouse and human sections, respectively.

Fig EV1F–F'"). The high expression of NR2F1 in the human and mouse NR, OD and ON suggests a conserved role for this gene in the development of the peripheral visual system. We observed that along the human ON, NR2F1 expression levels seem even higher than in mouse (compare Fig 1I–I" with Fig 1L–L"), suggesting an important role for NR2F1 in human ON development.

## Optic disc malformations in *Nr2f1* mutant mice

In light of the early *Nr2f1* expression gradient in the developing optic vesicle, key markers of proximo-distal (P-D) eye patterning, such as Pax6 and Pax2 (Schwarz *et al*, 2000), were first analyzed in *WT*, *HET,* and *KO* embryos. We found ectopic Pax6 expression in the presumptive ventral OS of *HET* and *KO* E10.5 embryos (Fig EV2A–F), as well as Pax2 downregulation and reciprocal Pax6 upregulation in invaginating E11.5 mutant optic cups (Fig EV2G–M"). Thus, differently from previous reports (Tang *et al*, 2010, 2015), loss of *Nr2f1* alone is sufficient to affect early patterning of the developing optic vesicle, indicating that Nr2f1 can control key molecular regulators of early identity acquisition during eye development.

The refinement of P-D marker expression is essential for the establishment of a structural border between OS and neural retina (NR), called the optic disc (OD), which forms at the most proximal region of the optic cup. In light of the ventral shift of the presumptive OS/NR border in *Nr2f1* mutants, and of BBSOA patients displaying OD abnormalities (Bosch *et al*, 2014; Chen *et al*, 2016), we decided to closely follow the morphological and molecular development of the OD in *Nr2f1* HET and KO eyes. Abnormal positioning and aberrant morphology of the presumptive OD was found in E12.5 *Nr2f1* mutants (Fig 2A–B"), particularly in *Nr2f1* KO optic vesicles, in which Pax2 expression is lost in the proximal OS region and ectopic Pax6$^+$ cells appear to differentiate into Tuj1$^+$ neurons *in loco* (arrowheads and inset in Fig 2C–D"). Cells positive for Pax2 remain abnormally low in the OD domain of *Nr2f1* HET and KO mutants until E18.5 (Fig 2E and F) and fail to properly surround and constrain the extension of Tuj1$^+$ axons already at E15.5 (Fig 2G–G'"). Early tissue patterning defects ultimately impinge on the final morphology and cell organization of the OD at later stages, in which Tuj1$^+$ fibers and Brn3a$^+$ RGC bodies remain severely misplaced in E18.5 *KO* fetuses (Fig 2H–I'). Together, our data show that reduced Nr2f1 expression levels affect P-D molecular patterning and lead to an abnormal OD organization from which RGC axons exit and form the ON. Interestingly, these defects are reminiscent of various OD abnormalities described in BBSOA patients, including small discs, pale discs, and disc excavations (Bosch *et al*, 2014; Chen *et al*, 2016).

## Pre- and early postnatal optic nerve defects in *Nr2f1* mutants

Since Nr2f1 promotes cell differentiation in different regions of the CNS (Faedo *et al*, 2008; Lodato *et al*, 2011a; Parisot *et al*, 2017; Bertacchi *et al*, 2018), we asked whether partial or complete loss of Nr2f1 function in the NR would affect cell proliferation and/or RGC differentiation (Fig 2J–V). Retinal neurons start to be generated around E11.5 in the mouse eye and express the neuron-specific β-tubulin marker Tuj1 and the RGC marker Brn3a (Heavner & Pevny, 2012). A reduction of Tuj1$^+$ differentiating cells in the ventral but not dorsal retina was detected in E13.5 *HET* and *KO* optic cups (Fig 2J–L), and consequently, fewer axons entered the OS (arrowheads in Fig 2K and K'). Reduced rates of differentiation were most probably due to an increased number of proliferative EdU$^+$/Ki67$^+$ cells resulting in tissue expansion and bending (arrowhead in Fig 2J' and Appendix Fig S1A–B'). Even if these morphological deformations might resemble a coloboma (Appendix Fig S1C–D'"),

which normally results from a failure in ventral fissure closure (Patel & Sowden, 2017), transverse sections of the optic vesicle depict proper fusion of the ventral fissures in E13.5 *Nr2f1* mutants (Appendix Fig S1E–J). Therefore, abnormal optic vesicle morphology observed in *Nr2f1* mutants arises most probably from tissue bending due to excessive proliferation and delayed cell differentiation.

A delay in RGC ventral differentiation was further confirmed by Brn3a staining, which results particularly reduced in malformed NR regions of E13.5 mutant embryos (Fig 2M–N'). However, *HET* and *KO* animals gradually recover their initial differentiation defect, since a comparable number of RGCs was quantified at E15.5 and E18.5 (Fig 2P). Accordingly, the diameter of the ON encompassing retinal fibers was only slightly reduced in *KO* animals at birth and not affected in *HETs* (P0; Fig 2Q–Q" and S). However, the number of RGCs decreased again in P7 *KO* and *HETs* (Fig 2P) and remained low at P28 (Fig 2O and P), in line with the affected ON size (Fig 2R and S; note that *KO* animals do not survive past P8). To assess whether RGC death could explain this postnatal defect, we stained *KO* and *HET* retinae with Brn3a and Caspase3, a marker for apoptotic cells. The number of apoptotic Brn3a$^+$ RGCs was increased in both E18.5 *HET* and *KO* retinae compared with *WT* (Fig 2T–U'), but only slightly changed at P5 and was no more detected at P7 (Fig 2V, M.S., M.B. unpublished observation). Together, our data show that reduced dosage of *Nr2f1* not only delays RGC differentiation prenatally, but also induces acute RGC death around birth resulting in progressive ON hypoplasia at early postnatal stages.

## Altered astroglia/oligodendroglia balance in *Nr2f1*-deficient optic nerves

The observation that ON atrophy worsens postnatally in mutant mice suggests that additional defects other than reduced RGC number and perinatal cell death might contribute to ON degeneration. Astrocytes and oligodendrocytes are two key glial populations supporting the stability and function of axonal fibers in the ON (Tsai & Miller, 2002; Hernandez *et al*, 2008). While primary astrocytes are mainly generated *in loco* from Pax2$^+$ cells of the OS inner layer and express NF1A and S100β (Tsai & Miller, 2002; Sun *et al*, 2017), mouse oligodendrocyte precursors originate from the preoptic area of the ventral forebrain and enter the ON at the chiasmal region around E18.5. They subsequently migrate, proliferate, and differentiate along the ON until P28, when myelination is accomplished (Ono *et al*, 2017). Oligodendrocytes can be labeled by Sox10 and myelin basic protein (MBP) expression. MBP is exclusive of fully differentiated oligodendrocytes that actively embed axonal fibers in myelin sheets to allow for efficient signal transmission (Stolt *et al*, 2002, 2004). NF1A and Sox10 mutually antagonize each other to control glial sub-lineage differentiation (Glasgow *et al*, 2014); hence, these markers are specific for their lineages (astrocytes versus oligodendrocytes, respectively) and virtually never overlap.

Since Nr2f1 is expressed at high levels in both astro- and oligodendrocyte cell populations of the developing ON (Figs 1 and EV1), we analyzed the effects of *Nr2f1* reduced dosage on the differentiation of both lineages at different developmental times. We found an increase of Pax2$^+$ astrocytic progenitors and NF1A$^+$ precursors in the inner layer of the OS in *Nr2f1* mutants at E13.5, and in the retinal, medial, and chiasmal regions of the ON at E15.5 and E18.5/P0

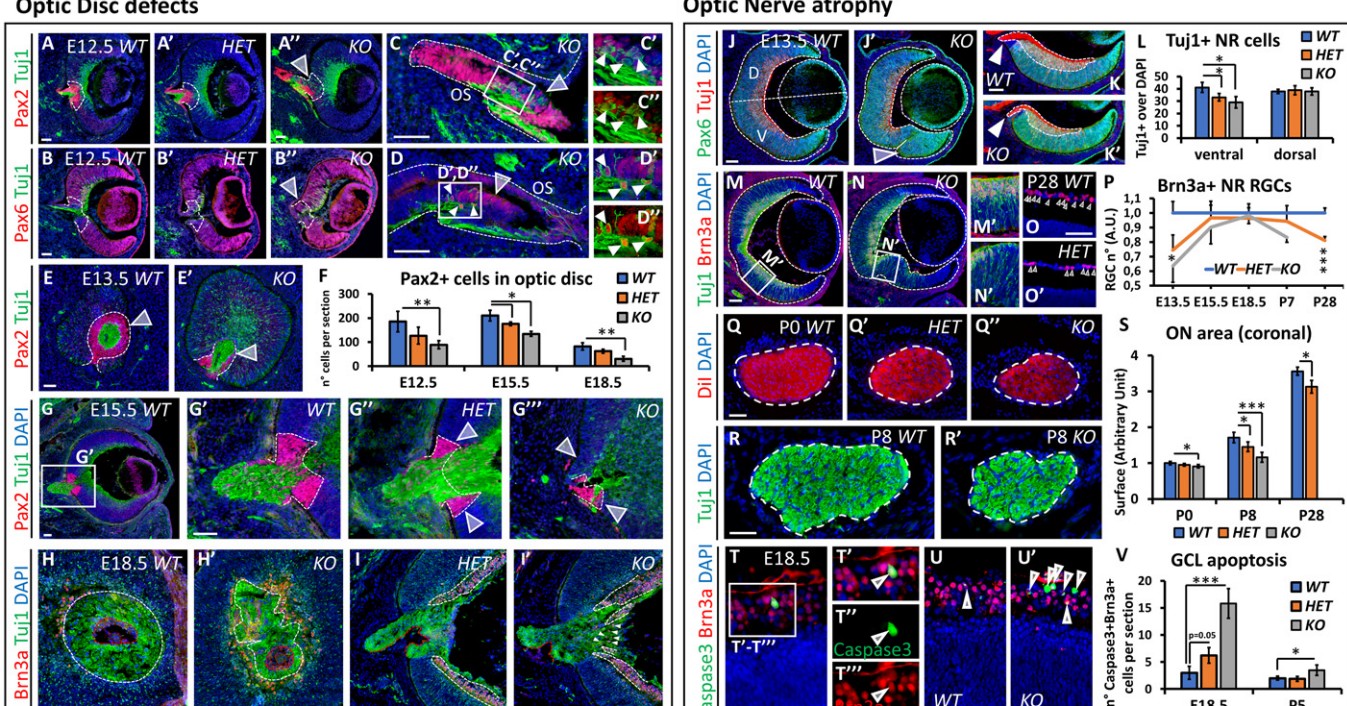

**Figure 2.  Optic disc (OD) malformations and optic nerve (ON) atrophy in *Nr2f1*-deficient mice.**

A–A″    Pax2 (red, OD) and TujI (green, axons) IF on E12.5 optic cup cross-sections in wild-type (*WT*), heterozygous (*HET*), and knock-out (*KO*) embryos showing abnormal ODs (white dotted lines and arrowhead).

B–B″    Pax6 (red, retinal progenitors) and Tuj1 (green) IF indicating ectopic Pax6⁺ retinal tissue in *HET* (B′) and *KO* embryos (B″, arrowhead).

C–D″    High-magnification views highlight a morphological displacement of Pax2⁻ (C–C″) and Pax6⁺ (D–D″) cells. Arrowheads in (D′, D″) point to ectopic Tuj1/Pax6⁺ neurons.

E, E′    Pax2 (red) and Tuj1 (green) IF on tangential sections of E13.5 optic cups revealing strong OD reduction in *KO* embryos (dotted line and arrowhead in E′), compared to *WT* (surrounded by a white dotted line in E).

F    Histogram quantifying the gradual reduction of Pax2⁺ OD cells in *HET* and *KO* at different ages.

G–G‴    Low (G) and high (G′–G‴) magnifications of E15.5 optic cups stained for Pax2 (red) and Tuj1 (green) confirming Pax2⁺ reduction in *HET* and *KO* ODs and severe morphological malformations in *KO* (arrowheads in G‴).

H–I′    Tangential (H, H′) and sagittal (I, I′) sections of the OD in E18.5 *WT* and *KO* retinae stained for Brn3a (red; RGCs) and Tuj1 (green; axons) highlighting the malformed OD and disorganized arrangement of RGCs in *KO*. In (H, H′), white dotted line delineates the borders of the OD and in (I, I′) the RGC layer; red dotted line surrounds mesodermal cells forming the hyaloid vessel. Arrowheads in (I′) point to ectopically placed RGCs.

J–K′    IF on E13.5 optic cup cross-sections in *WT* and *KO* embryos showing increased Pax6 (green, retinal progenitors) and decreased Tuj1⁺ (red, differentiating cells) in the ventral retina of *HET* and *KO* embryos. Arrowhead in (J′) points to increased thickness and bending of the ventral retina, while arrowheads in (K, K′) indicate Tuj1⁺ axons exiting the OD.

L    Histogram confirming the decreased number of Tuj1⁺ neurons in the ventral, but not dorsal, retina of *Nr2f1* mutants.

M–N′    Double Tuj1/Brn3a IF on E13.5 *WT* and *KO* optic cup sections revealing decreased numbers of differentiating RGCs in bending ventral retinae of *KO* embryos (N′).

O, O′    Details of P28 *WT* and *HET* retinae indicating decreased number of Brn3a⁺ neurons (arrowheads) in the GCL of *HET* animals.

P    Graph illustrating the dynamics of RGC differentiation from E13.5 to P28, normalized to *WT* animals.

Q–R    Cross-sections of (Q–Q″) DiI-labeled P0 ONs and (R, R′) Tuj1⁺ P8 ON fibers in *WT* and *KO* pups showing a progressive volume loss from P0 to P8.

S    Quantification of the surface occupied by Tuj1⁺ or DiI⁺ fibers in different genotypes and ages, as indicated.

T–U′    Cleaved Caspase3 (green, apoptotic cells) and Brn3a (red, RGCs) IF on transverse sections of E18.5 retinae in *WT* and *KO* fetuses. High magnifications (T′–T‴) highlight an example of a double-labeled RGC undergoing apoptosis (arrowheads) in the ganglion cell layer (GCL). Apoptotic RGCs are increased in *KO* animals (arrowheads in U′), compared to *WT* (U).

V    Histogram quantifying the number of RGC apoptotic cells in *WT*, *HET*, and *KO* from E18.5 to P5. Note the significant increase of dying cells at E18.5.

Data information: Nuclei (blue) were stained with DAPI. In (F, L, P, S, V), data are represented as mean ± SEM; *N* = 3–4 for (F, L, P, V); *N* = 4–5 for (S). Student's *t*-test (*$P < 0.05$, **$P < 0.01$, ***$P < 0.001$). Scale bars: 50 μm.

(Fig 3A–I). An increased NF1A⁺ astrocyte production is still maintained at P7 and leads to an over-represented astrocytic population in both *HET* and *Null* animals (Fig 3J–L). Strikingly, this abnormal astrocytic pool, intermingled between Tuj1⁺ fibers, shows a hypertrophic morphology (Fig 3M–O″). This is confirmed by the presence of expanded astrocytic processes containing large bundles of intermediate filaments in P8 *KO* transmission electron microscopic (TEM) thin sections (arrowheads and red dotted lines in Fig 3P and Q). Astrocytes appear to form a tight network and to display stress granules and a more heavily stained cytoplasm in mutants, not detected in *WT,* and compatible with a "reactive" state observed during neuroinflammation (Appendix Fig S2A–D′). Although there

was a clear difference between *WT* and mutant ON astrocytes prenatally, they seem to acquire a hypertrophic phenotype right around the first week of age, as described in other cases (Bovolenta *et al*, 1987). We thus assessed the presence of reactive astrocytes at P28, when their development was over, using Sox2, normally upregulated by pro-inflammatory signals triggering astrocytic proliferation (Bani-Yaghoub *et al*, 2006). P28 *Nr2f1 HET* ONs show a 25% increase of high-expressing Sox2$^+$ astrocytes with abnormal morphology compared with *WT* Sox2-expressing cells (Fig 3R–T). This was confirmed by TEM analysis illustrating enlarged astrocytic processes in *HET* ONs (Fig 3U and V). Interestingly, astrocyte inflammation was associated with abnormal mitochondrial morphology (Fig EV3), in line with a recently described mitochondrial involvement in BBSOA pathogenesis (Martin-Hernandez *et al*, 2018). At P8, astrocyte mitochondria become hypertrophic (1.78 ± 0.15 times larger than WT ones) with disorganized cristae, whereas RGC axonal mitochondria have a normal size (Fig EV3A–F). However, RGC somata show increased staining of the mitochondrial ATPase inhibitor ATPIF1 at later stages (Fig EV3G–H'), suggesting that mitochondrial biogenesis and function might be also altered in RGC cells. Together, these data show that reduced Nr2f1 dosage promotes an inflammatory process starting at early postnatal stages and possibly involving mitochondrial dysfunction.

### Miconazole treatment rescues improper oligodendrocyte differentiation and myelin lamination observed in *Nr2f1*-deficient mice

While the astrocytic population is over-represented in Nr2f1-deficient P8 ONs, the Sox10$^+$ oligodendrocyte population shows a dose-dependent opposite trend (Fig 4A–C), in a way that the glial population is imbalanced with astrocytes outnumbering oligodendrocytes (Fig 4C). Thus, we evaluated the proportion of proliferating, migrating, and differentiating oligodendrocytes by co-labeling Sox10$^+$ cells with the proliferative marker Ki67, the apoptotic marker Caspase3, and the terminal differentiation protein MBP in normal and mutant ONs (Fig EV4). While there is no difference in proliferation or apoptosis between *Nr2f1* mutants and *WT* littermates, the ratio of undifferentiated (Sox10$^+$/MBP$^-$) versus differentiated (Sox10$^+$/MBP$^+$) oligodendrocytes is significantly increased in *HET* and *KO Nr2f1* mutant ONs (Fig EV4A–I), indicating failed oligodendrocyte maturation upon reduced *Nr2f1* expression. More Sox10$^+$ cells are present in the proximal versus distal ON portions in both *HET* and *KO* animals (Fig EV4H), suggesting also defective oligodendrocyte migration along the nerve. Thus, deficiency in *Nr2f1* genetic dosage affects migration and differentiation of developing oligodendrocytes (summarized in Fig EV4J).

Next, we asked whether impaired oligodendrocyte development would affect fiber myelination using MBP as a late marker. At P8, the proximal and distal ON portions of both *HET* and *KO* animals show a drastic reduction of MBP immunostaining compared with *WT* (Fig 4D–J). Notably, *Nr2f1 HET* mice have impaired myelination at the proximal end to an extent similar to that of *KO* animals (Fig 4E, F and J), indicating that fine regulation of gene dosage is essential for proper oligodendrocyte differentiation. TEM analysis confirmed a lower amount of myelinated fibers in both *HET* and *KO* ONs, although with different severity (Fig 4K–N). The few

myelinated fibers in P8 *KO* animals display thin and incomplete sheets (Fig 4M, inset). Poor myelination is still maintained in P28 *HET* animals (Fig 4O–Q), thus excluding a transient defect. Indeed, high-magnification EM images show that several *HET* fibers are surrounded by disorganized and poorly compacted myelin sheaths (g-Ratio: 0.63 ± 0.04), when compared with the dense myelin layer of *WT* animals (g-Ratio: 0.74 ± 0.02; Fig 4R and S). Together, our findings show that *Nr2f1*-deficient oligodendrocytes fail to properly migrate and differentiate during development, ultimately leading to reduction of myelin surrounding optic nerve fibers in juvenile mutant mice.

Miconazole and Clobetasol act as two potent inducers of oligodendrocyte differentiation, as they trigger fast maturation of MBP$^+$ cells in both physiological and pathological conditions (Najm *et al*, 2015; Su *et al*, 2018). Given our observations, we asked whether a similar treatment could be of use in the BBSOA mouse model of optic neuropathy. Using previous protocols (Najm *et al*, 2015), we injected Miconazole or Clobetasol daily in control and *Nr2f1 HET* pups from P2 to P8 (Fig 5A). Miconazole, but not Clobetasol, which resulted toxic for the pups (M.B., M.S. unpublished observations), increased the number of Sox10$^+$ precursors in *Nr2f1 HET* ONs and improved fiber myelination to a level comparable to that of *WT* littermates (Fig 5B–E). Miconazole has been showed to specifically enhance ERK1/2 activity (Najm *et al*, 2015). Consistently, the phosphorylated (i.e., activated) form of ERK is upregulated upon Miconazole treatment (Fig 5F and G), as quantified in whole ON (Fig 5H) and specifically in the cytoplasm of Sox10$^+$ oligodendrocytes (Fig 5I). To understand whether the short Miconazole treatment was enough to sustain fiber myelination even at later stages, we analyzed P2- to P8-treated animals at P28 (Appendix Fig S3A–H). Notably, treated ONs maintain oligodendrocyte number and myelination recovery at later postnatal stages (Appendix Fig S3A–H). Furthermore, myelin compaction is partially rescued by Miconazole treatment (Fig 5J–M). However and interestingly, the number of Sox2$^+$ reactive astrocytes and their aberrant morphology are not modified in treated pups and remain similar to untreated ones (Fig 5N–Q), strongly suggesting that astrocyte alterations are not secondary to myelination defects in *Nr2f1*-deficient animals.

### Thalamic and cortical visual regions are reduced in *Nr2f1*-deficient mice

Since Nr2f1 is also expressed in the visual thalamus and cortex (Armentano *et al*, 2006; Chou *et al*, 2013), we wondered whether the thalamic relay of RGC visual fibers, the dorsal lateral geniculate nucleus (dLGN), was altered in *Nr2f1*-deficient animals (Fig EV5). At both P0 to P8, the size of the dLGN, measured in Nissl-stained sections, was progressively and significantly reduced in both *HET* and *KO* animals when compared with *WT* (Fig EV5A–F), as also reported in thalamic-specific conditional *Nr2f1 KO*s (Chou *et al*, 2013). Furthermore, and possibly related to the reduction of both the RGC number and the dLGN size, DiI-labeled E18.5 RGC axons were less efficient in invading the dLGN as compared with their *WT* counterparts (Fig EV5G–I). Moreover, since *Nr2f1* is one of the major genes orchestrating cortical arealization (Zhou *et al*, 2001; Armentano *et al*, 2007; Alfano *et al*, 2013), we wondered whether reduced *Nr2f1* expression would also affect the size of neocortical

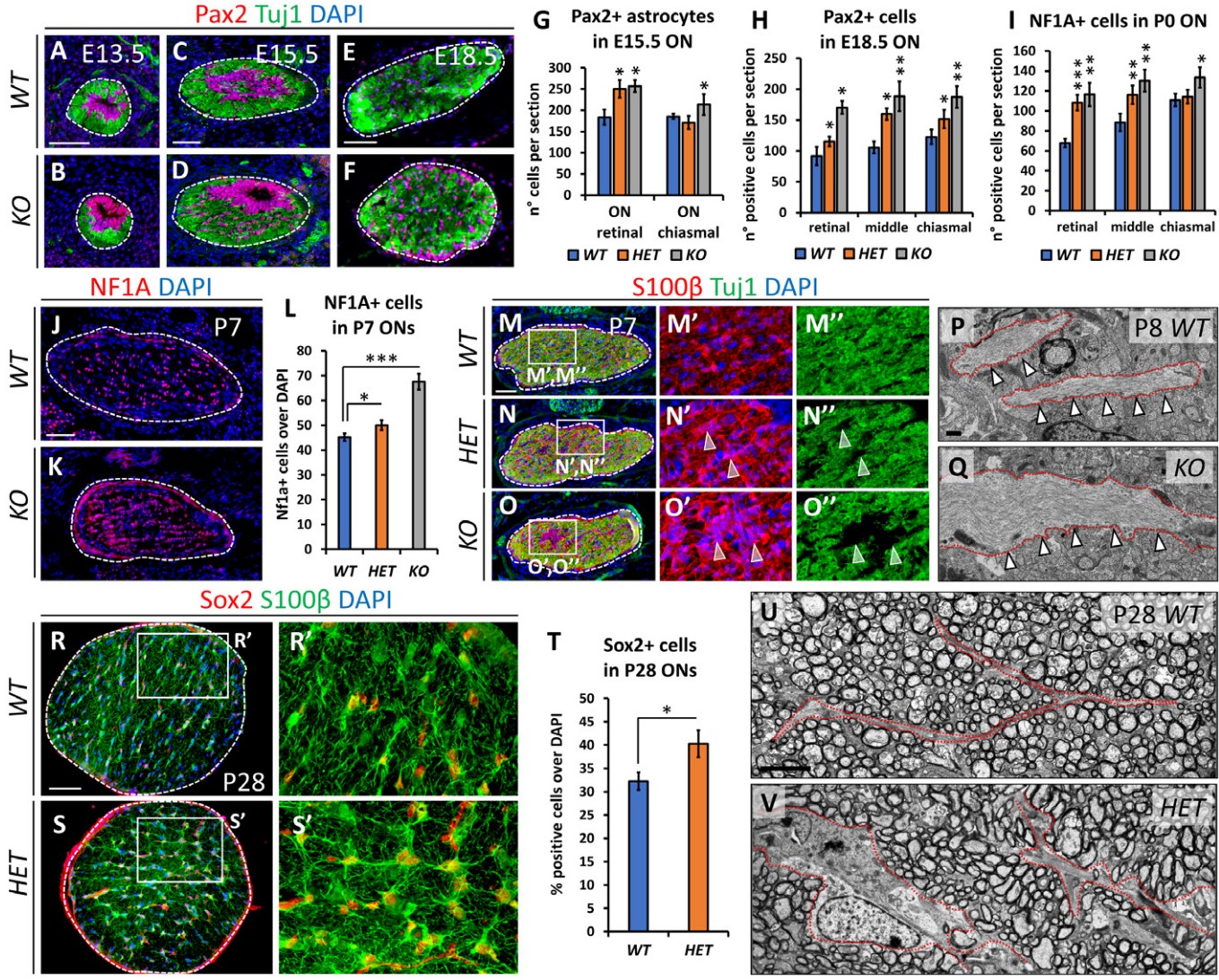

**Figure 3. Increased astrocytic population in *Nr2f1*-deficient optic nerves.**

A–F    Pax2 (magenta, astrocytic progenitors) and Tuj1 (green, RGC axons) IF on E13.5, E15.5, and E18.5 OS/ONs cross-sections showing an increase of Pax2-expressing cells within the optic nerve (ON) of *KO* embryos at all stages.

G–I    Histograms quantifying the average number of Pax2+ (G,H) or NF1A+ (I) cells in *WT*, *HET*, and *KO* at different stages and axial length of the ON, as indicated. Astrocytic progenitors are significantly increased along the nerve of mutants.

J, K    NF1A (red, astrocyte precursors) IF on P7 *WT* and *KO* ON cross-sections illustrating a high astrocytic density in *KO* ON (delineated by thin dashed lines).

L    Histogram confirming the increased percentage of NF1A+ astrocytes in *HET* and *KO* ONs compared to *WT*.

M–O"    Tuj1 (green, RGC axons) and S100β (red, astrocytes) IF on P7 *WT*, *HET*, and *KO* ONs. Arrowheads point to S100β+ astrocyte groups (N', O') and absence of Tuj1+ fibers (N", O") in *HET* and *KO* ONs.

P, Q    TEM images of *WT* and *KO* P8 ONs showing thick astrocytic processes filled with intermediate filaments (red dotted lines and arrowheads) in *KO* nerves.

R–S'    S100β (green, astrocytes) and Sox2 (red, reactive/inflamed astrocytes) IF on P28 *WT* and *HET* ONs illustrating strong Sox2 expression in reactive astrocytes.

T    Graph quantifying the average number of Sox2+ astrocytes per P28 ON section in *KO* and *WT* animals.

U, V    EM sections of P28 *WT* and *HET* ONs illustrating the expanded astrocytic processes (surrounded by red dotted lines) among myelinated fibers of P28 ONs.

Data information: In (G, H, I, L, T), data are represented as means ± SEM; $N = 3$–5. Statistical significance was obtained by Student's *t*-test (*$P < 0.05$; **$P < 0.01$; ***$P < 0.001$). Nuclei (blue) were stained with DAPI. Scale bars: 50 μm, except (P, Q) (500 nm) and (U, V) (4 μm).

areas, particularly sensory areas (including the visual cortex). Consistently, the *Lmo4*-expressing visual cortex of *HET* animals is slightly reduced, even if not as severely as observed in *KO* brains (Appendix Fig S4A–D). Together, our data indicate that *Nr2f1* haploinsufficiency affects the development of the entire visual system, from the retina to the visual cortex.

**Reduced axonal conduction velocity in the visual pathway of *Nr2f1* mutant adults**

To test whether anatomical alterations in the *Nr2f1* heterozygous visual system had functional consequences, we quantified the conduction velocity of nerve impulses along the visual pathway by implanting

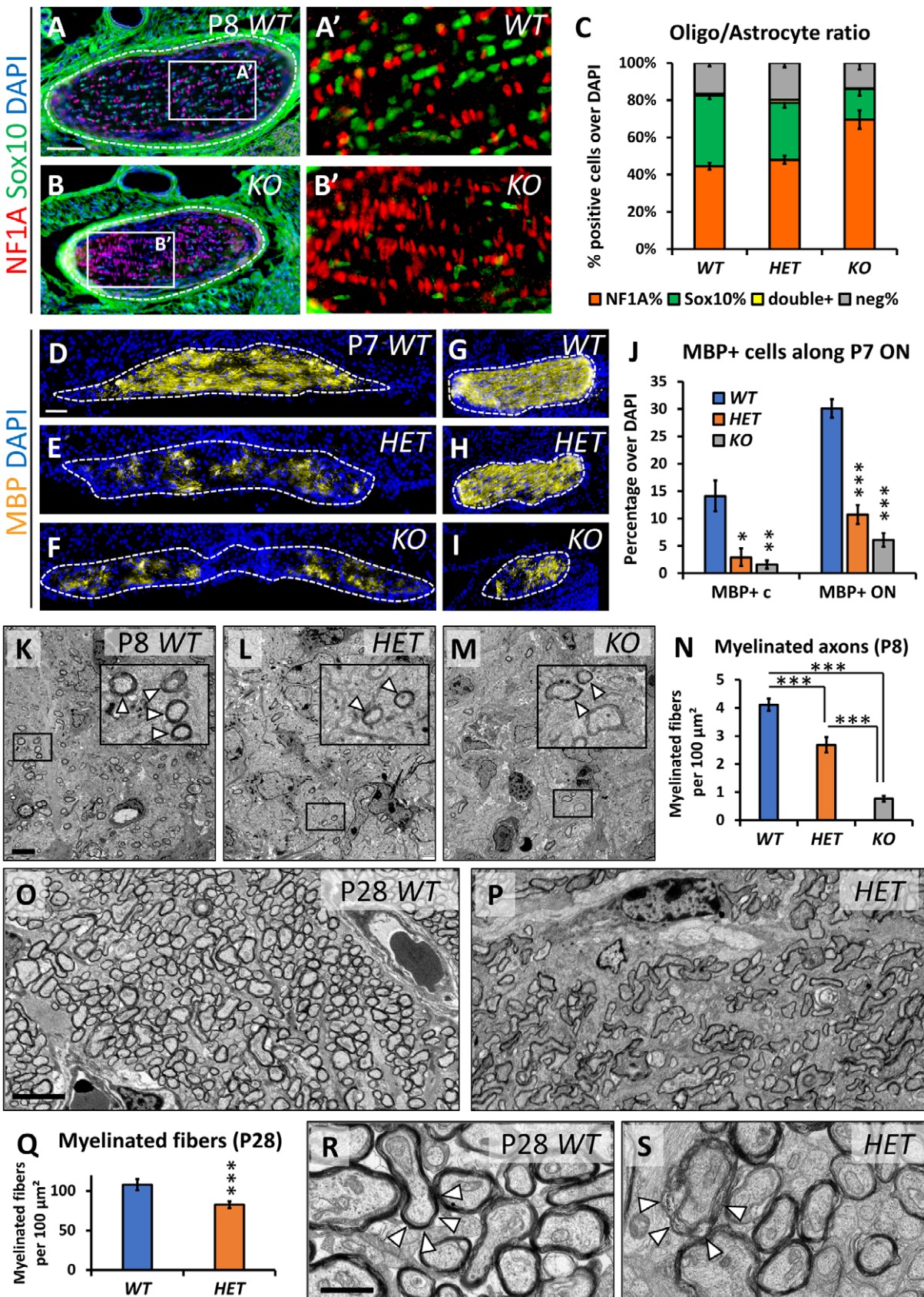

**Figure 4.**

**Figure 4.  Optic nerve myelination is affected by reduced *Nr2f1* dosage.**

A–C   NF1A (red, astrocytes) and Sox10 (green, oligodendrocytes) IF on cross-sections of *WT* and *KO* P8 ONs showing altered ratio of astrocytes and oligodendrocytes in mutants, particularly in *KO*s, as quantified in (C).

D–J   MBP (yellow, fully differentiated oligodendrocytes) IF on cross-sections of *WT*, *HET*, and *KO* P7 optic nerves (ONs) at the chiasm (D–F) and between the chiasm and the eye (G–I). Note the dramatic decrease of MBP$^+$ oligodendrocytes at the chiasm in *HET* and *KO*, quantified in (J). The ratio of MBP$^+$ cells is calculated over the total DAPI$^+$ cell number per chiasmal/ON section.

K–N   EM thin sections of P8 *WT*, *HET*, and *KO* ONs depicting myelin as a dense, dark staining around axonal fibers illustrate a significantly lower number of myelinated fibers in *HET* and *KO*, as quantified in (N). Higher magnification images (insets) showing different degrees of myelination in *HET* and *KO* ONs. Arrowheads in insets point to myelinated fibers.

O–Q   TEM images of P28 *WT* and *HET* confirming persistent hypomyelination in mutants at P28, as quantified in (Q).

R, S   High-magnification EM images of P28 *WT* and *HET* ONs showing the difference in myelin compaction between *WT* (arrowheads in R) and *HET* (arrowheads in S).

Data information: In (C, J, N, Q), data are represented as means ± SEM. $N$ = 4–5 for (C, J); $N$ = 3 for (N, Q). Statistical significance was obtained by Student's $t$-test or by two-way ANOVA when comparing 2 or multiple conditions, respectively (*$P$ < 0.05; **$P$ < 0.01; ***$P$ < 0.001). Nuclei (blue) were stained with DAPI. Scale bars: 50 μm, except (K–P) (4 μm) and (R, S) (1 μm).

recording electrodes into their dLGN, superior colliculus, and visual cortex (Fig 6A and B). Alert-behaving *WT* and *HET* adults were stimulated with light flashes presented bilaterally 40 times per session at a rate of 6/min. As previously described (Wiggins *et al*, 1982; Meeren *et al*, 1998; Sanz-Rodriguez *et al*, 2018), flash stimulations evoke an early positive–negative–positive field potential followed by some late oscillatory components in the dLGN (Fig 6C). The latency values for the first positive component in *WT* mice are in line with reported mouse field potentials and unitary recordings evoked by photic stimulation (Lintas *et al*, 2013; Sanz-Rodriguez *et al*, 2018). Notably, *HET* animals display a significantly ($t$ = −6.385 with 22 degrees of freedom; $P$ < 0.001) longer mean conduction velocity (15.79 ± 048 ms) for the first component of the evoked field potential than *WT* mice (12.38 ± 0.25 ms; Fig 6C and D), suggesting decreased transmission of retinal impulses upon *Nr2f1* reduced dosage. At the level of the superior colliculus, field potentials present a succession of negative-positive components that last for > 80 ms. In this case, the latency to the initiation of the first negative component is also significantly longer ($t$ = −2.148 with 18 degrees of freedom; $P$ = 0.046) in *Nr2f1 HET* (17.60 ± 0.58 ms) than in *WT* (15.95 ± 0.50 ms) mice (Fig 6C and D). Finally, field potentials evoked in the primary visual cortex produce an early and noticeable positive–negative–positive component followed for a long-lasting (> 80 ms) succession of negative waves sometimes riding on the top of a slow positive wave (Fig 6C; Wiggins *et al*, 1982; Meeren *et al*, 1998; Ridder & Nusinowitz, 2006). The latency to the first positive component is again significantly longer ($t$ = −3.073 with 26 degrees of freedom; $P$ = 0.005) for *Nr2f1 HET* (16.75 ± 0.41 ms) than for *WT* (14.64 ± 0.55 ms) mice (Fig 6D). Taken together, these data show that *Nr2f1*-deficient mice exhibit a significant defect in the transmission of visual stimuli from the retina to the visual cortex. Since these differences were similar (≈2 ms) among the three recording sites, we can assume that the deficits in axonal conduction velocity are mainly restricted to the optic nerve.

**Visual and associative learning deficits in *Nr2f1* heterozygous mice**

BBSOA patients have been diagnosed with cerebral visual impairments (CVI; Bosch *et al*, 2014; Chen *et al*, 2016), a condition in which visual stimuli are not properly interpreted and associated, and which relates to damage or malfunction of the visual pathway or visual centers in the brain. To understand whether *Nr2f1* heterozygotes encountered similar associative visual and learning disorders, we tested their performance firstly in a light-dependent operant conditioning task

(Fig 6E, top panel) and later in the same box, but with a cue depending on a small light bulb switched on or off (Fig 6E, bottom panel). *WT* ($n$ = 13) and *Nr2f1* heterozygous (*HET*; $n$ = 18) mice were trained in a Skinner box to obtain a food pellet every time they press a lever in 10 daily sessions of 20 min each. In this preliminary training, the two groups reached the selected criterion (Fig 6F) at the same time ($U$ = 142.000; $P$ = 0.703; Mann–Whitney rank sum test). In addition, they performed similar numbers of lever presses ($t$ = −0.733 with 18 degrees of freedom; $P$ = 0.473; Fig 6G). After this pre-training, mice were subjected to a more complex task, where pressing the lever was only rewarded with a pellet during those periods (20 s) in which a small light bulb located above the lever was switched on (light on/light off protocol, Fig 6E, bottom panel). In this case, the *WT* group reached the selected criterion by the sixth conditioning session, while *Nr2f1 HET*s were unsuccessful in completing this task (Fig 6H). Values collected for the light on/light off coefficient (see Materials and Methods) were significantly higher for *WT* mice compared with *HET*s [$F_{(1,135)}$ = 6.129; $P$ = 0.026]. A best linear fit to the collected data points indicates that *WT* mice increased their performance across training (slope = 0.0632; $r$ = 0.75; $P$ = 0.046), whereas *HET* animals failed to show any sign of improvement (slope = −0.0118; $r$ = 0.199; $P$ = 0.809; Fig 6H). A further analysis indicated that *Nr2f1 HET* mice were unaware of the working code represented by the light bulb. Indeed, and as shown in Fig 6I, while *WT* mice learned to press the lever preferentially ($t$ = 5.272 with 24 degrees of freedom; $P$ < 0.001) during the light on (≈75%) versus the light off (≈25%) periods, *HET*s pressed the lever equally (≈50% each) during these two periods ($t$ = 0.091 with 16 degrees of freedom; $P$ = 0.929). In conclusion, *Nr2f1* heterozygous mice show a significant deficit in associating the dim visual cue with the operant task. Although the negative slope value of the regression line illustrated in Fig 6H suggests the possibility of a learning deficit, not necessarily related to visual limitations, the fact that *HET*s acquired the initial training in a well-illuminated box similarly to controls support the presence of a relevant deficit in the acquisition/processing of visual cues.

## Discussion

### BBSOAS, a new pathogenic model for optic atrophy

Dominant optic atrophy is the most common form of autosomally inherited (non-glaucomatous) optic neuropathy characterized by optic nerve pallor and reduced visual acuity. RGC alterations have

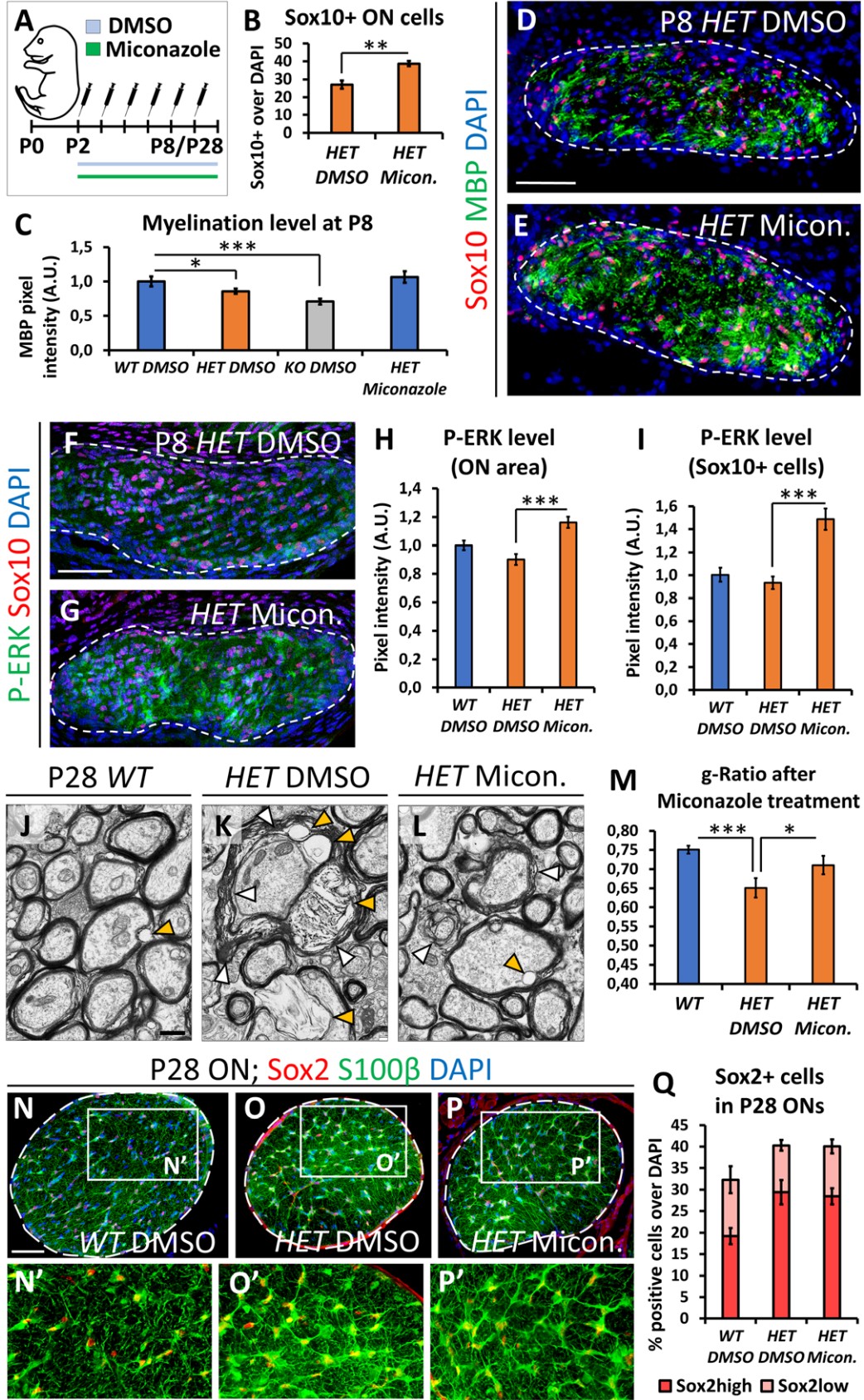

Figure 5.

◀

**Figure 5.** **Miconazole treatment specifically rescues Nr2f1-dependent hypomyelination in postnatal pups.**

A    Experimental design to treat *HET* pups from P2 to P8 with Miconazole. DMSO (controls) and Miconazole-treated pups were sacrificed at P8 or at P28 to evaluate the ON myelination.

B–E   Sox10 (red, oligodendrocyte precursors) and MBP (green, fully differentiated oligodendrocytes) IF on cross-sections of P8 *HET* ONs after 6 days of treatment with DMSO (D) or Miconazole (E). Miconazole increased the number of Sox10$^+$ cells compared to *HET*s treated with DMSO (quantified in B) and rescued MBP staining pixel intensity of *HET* pups to level comparable to those of *WT* pups (quantified in C).

F–I   Sox10 (oligodendrocyte marker, red) and Phospho-ERK (dually phosphorylated forms of active ERK1 and ERK2, green) IF in *WT* and *HET* ONs, treated with DMSO (F) or Miconazole (G) between P2 and P8. As previously reported, Miconazole strongly activate ERK signaling pathway (G). Signal intensity quantification in whole ON area is shown in (H), while specific evaluation of pixel intensity around Sox10$^+$ nuclei is shown in (I).

J–M   EM thin sections of P28 *WT* and *HET* ONs, treated with DMSO (J, K) or Miconazole (L) between P2 and P8. Different degrees of myelin compaction can be appreciated at high magnification, with *HET* ONs showing non-compacted sheaths (white arrows in K) and vacuoles (yellow arrows). Miconazole treatment rescues almost normal myelin g-ratio, as quantified in (M).

N–P'  S100β (green, astrocytes) and Sox2 (red, reactive/proliferative astrocytes) IF in the ONs of *WT* (N, N') and *HET* (O–P') animals after 6 days of treatment with DMSO (N–O') or Miconazole (P, P'), then sacrificed at P28. Astrocytes undergo remodeling and express Sox2 in *Nr2f1 HET* animals (O', P'), even after Miconazole treatment.

Q    Histogram showing the average number of S100β$^+$/Sox2$^+$ astrocytes per P28 ON section in *WT* or *HET* animals treated with DMSO or Miconazole as indicated, with Sox2 high (magenta columns) and low levels (pink columns).

Data information: Nuclei (blue) were stained with DAPI. In (B, C, H, I, M, Q), the error bars represent the SEM of the means; N = 3 for (B, C); N = 2 for (H, I, M, Q). Statistical significance was obtained by ANOVA (*P < 0.05; **P < 0.01; ***P < 0.001). Scale bars: 50 μm, except (J–L) (500 nm).

been proposed as the main cause of optic neuropathy. RGCs are long projection neurons with an initial long unmyelinated intraretinal segment that is thought to be particularly vulnerable to insults, at least in cases of LHON and OPA1 mutations, the major forms of optic atrophy described so far (Lenaers *et al*, 2012; Chun & Rizzo, 2017; Jurkute & Yu-Wai-Man, 2017). The underlying mechanisms that mediate RGC death and progressive ON degeneration are still not completely elucidated in the various forms of neuropathies. Our study now offers different mechanisms to explain optic atrophies of the BBSOA type (schematized in Fig 7).

First, *Nr2f1* heterozygous mice, analogous to *NR2F1* haploinsufficient patients, recapitulate important features of BBSOA patients, such as OD malformations and ON atrophy, and could serve as a model to study the pathogenesis of optic neuropathies and visual impairments. Furthermore and in contrast to what previously reported (Tang *et al*, 2010, 2015), we demonstrate that *Nr2f1* deletion alone is sufficient to induce major optic abnormalities in mice that, as in humans, do not seem to be compensated by its homolog *Nr2f2*. This contention is also supported by our novel observation that the expression pattern of *NR2F1* is strongly conserved between human and mouse, at least in the developing eye. In humans, *NR2F1* point mutations cause visual deficits with high penetrance, apparently in the absence of any additional gene mutation, indicating that our mouse model can be used to unravel the molecular/cellular mechanisms leading to human BSSOA syndrome, and more generally to understand the pathophysiology of ON neuropathy and/or degeneration. Why our findings differ from those reported by Tang *et al* (2012, 2015) is unclear; however, the most likely explanation reside in the use of a conditional mutant (*Rax-Cre Nr2f1*), in which the removal of the gene occurs either too late or less efficiently than in our model based on constitutive gene loss. We thus consider that mice heterozygous for *Nr2f1* well replicate the pathology of patients haploinsufficient for *NR2F1*.

Second, Nr2f1 seems to directly act on the migration and differentiation of oligodendrocytes and ultimately on the myelination degree of the ON. Oligodendrocyte precursors appear in the preoptic area of the ventral telencephalon around E12.5 in the mouse and migrate caudally toward the ON, where they form myelin sheaths wrapping axons (Ono *et al*, 2017). *Nr2f1* is highly expressed in the preoptic area (Armentano *et al*, 2006; Lodato *et al*, 2011b) and co-

localizes with Sox10 in ON oligodendrocytes, suggesting that *Nr2f1* expression follows oligodendrocyte development from its source to its target. Our data show a progressively reduced number of Sox10$^+$ from the proximal to the distal ON portions in both *Nr2f1 HET* and *KO* mutants indicating a delay of migrating oligodendrocytes in reaching more distal parts of the ON. This is in accordance with ON hypomyelination at P8, although with different severities between *HET* and *KO*, and in *HET* mutants at P28. Strikingly, postnatal treatment with Miconazole, a potent inducer of oligodendrocyte differentiation in several brain regions (Najm *et al*, 2015; Su *et al*, 2018), efficiently rescues the myelination grade of *Nr2f1* mutants via *ERK* signaling activation, showing that this chemical drug can counteract hypomyelination in the context of the BBSOA mouse model. This is a first step not only in understanding the pathophysiological features of a complex syndrome, but also in proposing possible therapeutic approaches. If proved efficient and safe in humans, Miconazole could be used in the future as a treatment to trigger oligodendrocyte maturation/differentiation and eventually improve ON signal conductance in BBSOA patients.

Third, our data show increased astrogliosis in the postnatal ON of *Nr2f1*-deficient mice. Differently from late-generated cells such as oligodendrocytes, astrocyte precursors are already present in the OS and ON at early stages and constitute the principal glial population at E18.5. Later on, they still represent the major glial cell type in the non-myelinated ON heads in most mammals (Hernandez *et al*, 2008) and respond to damage by changing from a quiescent to an activated state. During development, Pax2$^+$ astrocyte precursor cells express guidance molecules at the OD through which RGC axons exit (Chu *et al*, 2001). At later stages, NF1A- and S100β-expressing astrocytes are intermingled with oligodendrocytes along the ON providing structural support and nutrients to axons, as well as forming a scaffold around endothelial cells to help maintaining the integrity of the blood–brain barrier. Astrocytes become reactive and re-express the progenitor marker Sox2 in response to a range of pathological conditions, from acute traumatic injury to chronic neurodegenerative diseases (Bani-Yaghoub *et al*, 2006). In the eye, glial cell proliferation and/or activation (gliosis) usually occur as a reactive change associated with ON degeneration and/or glaucoma (Hernandez *et al*, 2008; Pekny *et al*, 2014). Reactive astrocytes are generally identified by morphological changes (hypertrophy, thicker

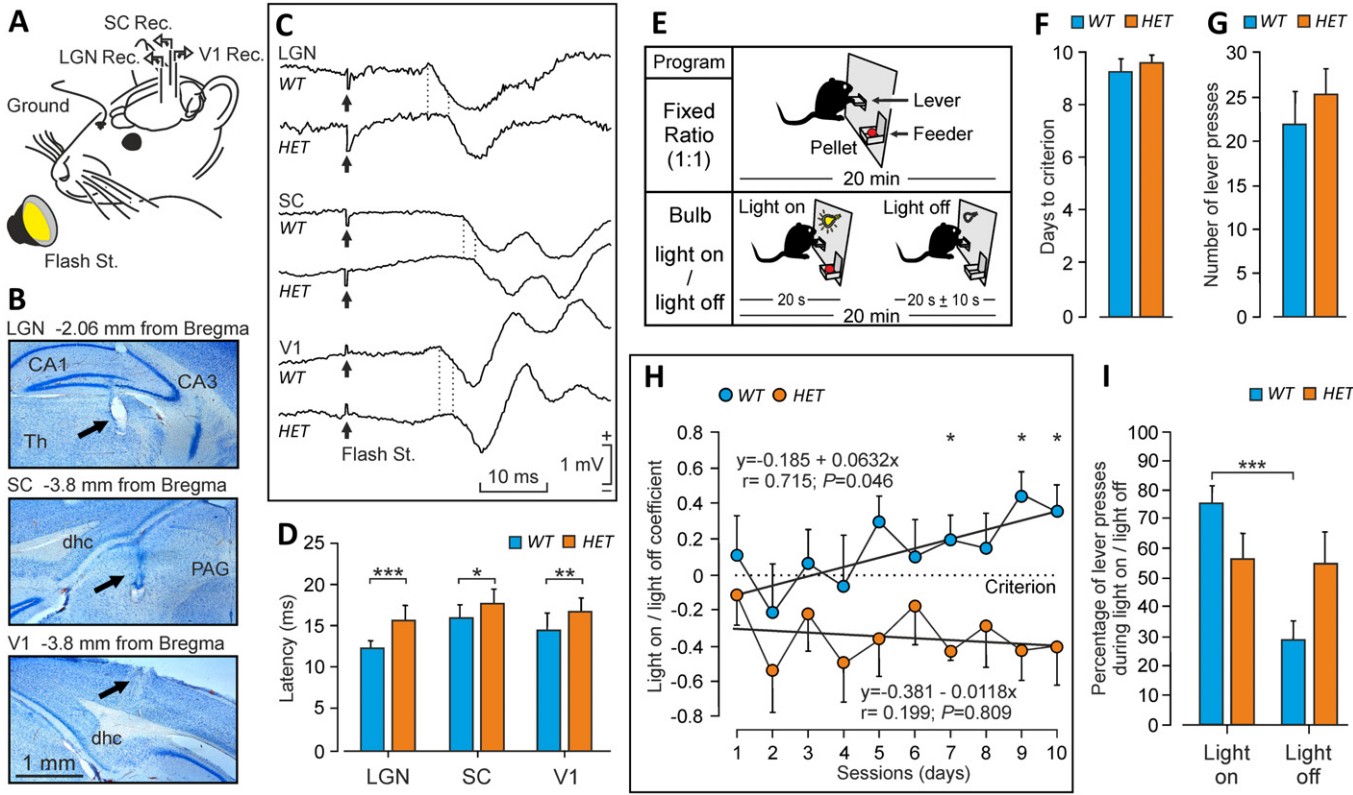

**Figure 6. Axonal conduction velocity and visually-dependent operant conditioning are impaired in *Nr2f1* HET mice.**

A  Animals were implanted with recording electrodes in lateral geniculate nucleus (LGN), superior colliculus (SC), and visual cortex (V1) and stimulated with a flashing stroboscope.

B  Nissl-stained sections illustrating the location of the recording electrodes (arrows). dhc, dorsal hippocampal commissure; PAG, periaqueductal gray; Th, thalamus.

C  Examples of field potentials (averaged ≥ 20 times) evoked by flash stimulation in the recording sites of *WT* and *HET* mice. Dotted lines indicate the starting point of evoked field potentials.

D  Latency of evoked field potentials for *WT* (N = 13) and *HET* (N = 18) mice.

E  In a first series of experiments, animals [*WT* (N = 13) and *HET* (N = 18)] were trained to press a lever to obtain a pellet of food in an illuminated Skinner box with a fixed-ratio (1:1) schedule. After 10 days of training, they were transferred to a light on/light off protocol where lever presses were only rewarded when a small light bulb located over the lever was switched on. Lever presses while the bulb was off were punished with a time penalty of ≤ 10 s during which the bulb would not turn on.

F, G  Mean days to criterion (F) and mean number of lever presses during the last two training sessions (G) for the fixed-ratio (1:1) paradigm. No significant differences were observed.

H  Performance of *WT* (N = 13) and *HET* (N = 18) mice in the light on/light off test. Criterion (dotted line) was that the animal had to press the lever more times during the small bulb light on period than during the light off one. *WT* mice performed better than *HET* animals and reached the criterion by the 6th session. The *HET* failed to reach the criterion for the duration of the test.

I  The percentage of lever presses during the light on and light off periods for *WT* and *HET* mice was also different, because the *WT* group pressed the lever preferentially during the light on period. Data were averaged from the 9th and 10th sessions. For statistical significance, see Materials and Methods (*P < 0.05; **P < 0.01; ***P < 0.001).

cell processes, larger and more vesicular nuclei, and in this case abnormal mitochondria) and tend to form a mesh (or glial scar) that alters the microenvironment of fibers, likely contributing to Wallerian degeneration and thus RGC cell death (Sofroniew, 2009; Conforti et al, 2014).

We found that Nr2f1 protein is expressed in the astrocytic population at early stages in mice as well as in humans, and that already at E13.5 the Pax2$^+$ astrocyte precursors are over-represented along the ON of mutant embryos. Abnormally high production of astrocytes continues at later stages in both *HET* and *KO* animals, suggesting that Nr2f1 might play an intrinsic role in modulating the

generation of astrocytes during ON development. This is supported by the finding that Nr2f1 cell-intrinsically drives progenitors toward neurogenesis by repressing astrogliogenesis in adult neural stem cells of the hippocampus (Bonzano et al, 2018). In its absence, neurogenesis is reduced at the expense of astrocyte production and, conversely, high Nr2f1 protein levels can rescue the hippocampal neurogenesis–astrogliogenesis imbalance due to neuroinflammation (Bonzano et al, 2018). Thus, Nr2f1 might finely modulate astrocyte production/proliferation not only during adult neurogenesis, but also, as shown in this study, during ON development. We propose that the appearance of reactive astrocytes along the ON at P8 and

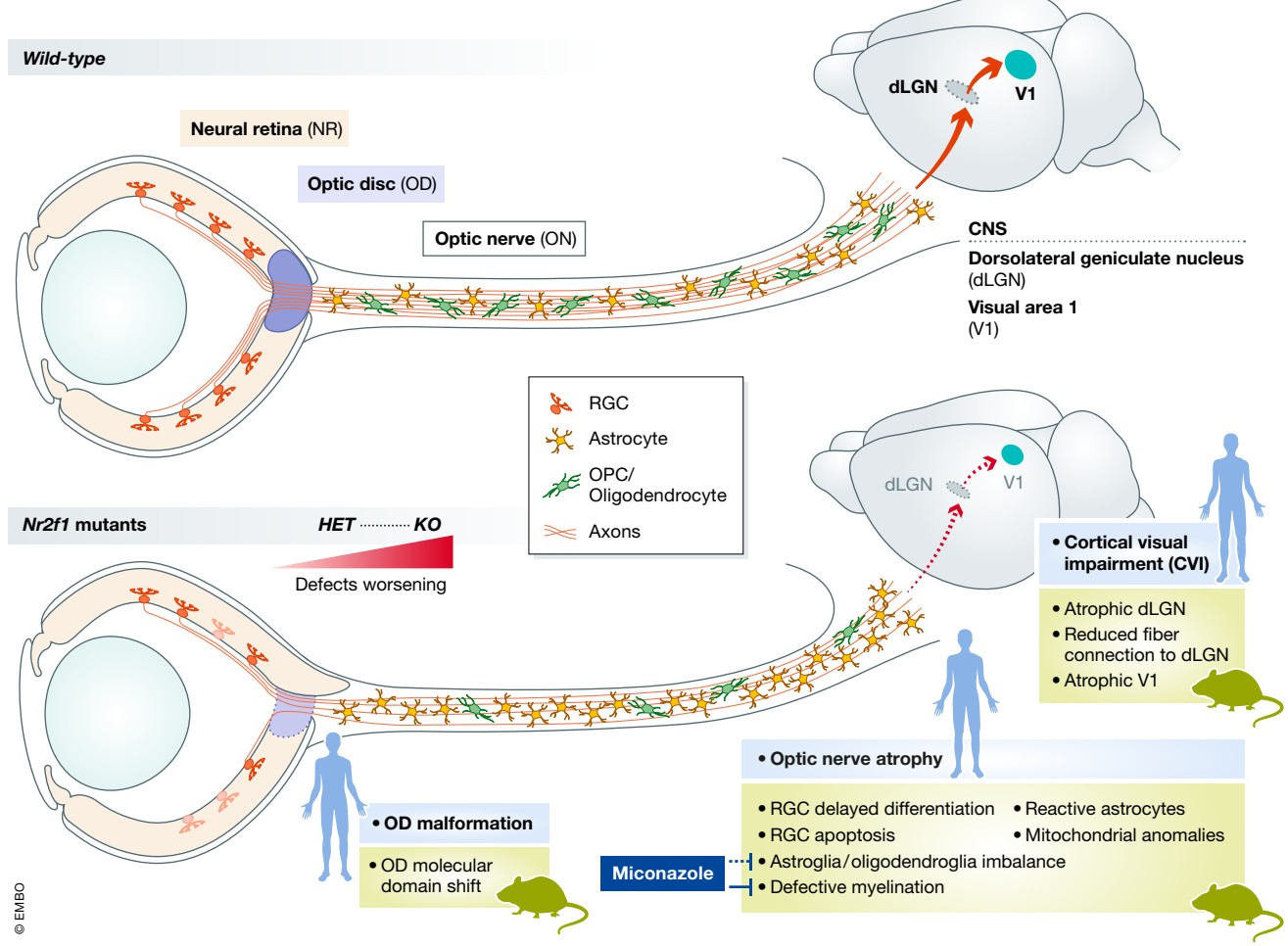

**Figure 7. Summary of the defects observed in the visual system of *Nr2f1* mutant mice compared to patient features.**

Schematic overview of the different phenotypes observed in the retina, optic nerve, thalamus, and visual cortex of *Nr2f1* heterozygous (*HET*) and knock-out (*KO*) mice. Some of the defects characterized in the mouse model (green boxes) are reminiscent of the major features reported in BBSOA patients (light blue boxes). *Nr2f1* haploinsufficiency causes a shift of neural retina/optic stalk molecular domains (in blue), leading to aberrant positioning of the optic stalk and ultimately optic disc (OD) malformations at early stages. Pale and abnormal ODs have also been reported in human patients. Optic nerve (ON) atrophy, the most characteristic feature of BBSOA patients, is most probably caused by: (i) delayed retinal ganglion cell (RGC) differentiation (red cells) at early embryonic stages and increased apoptosis at perinatal stages; (ii) affected oligodendrocyte precursor cell (OPC) migration and maturation (green cells) leading to ON fiber demyelination and reduced volume; (iii) reduced axonal innervation and connectivity to the atrophic thalamic dLGN and V1 neocortex; (iv) reactive gliosis consistent with astrocyte inflammation; and (v) possible associated mitochondrial dysfunctions. All these defects identified in *Nr2f1* mutants contribute to form a reduced and hypomyelinated ON and ultimately affect visual axonal conduction velocity. While the myelination defect could be rescued by Miconazole treatment dark (dark blue box), the high proportion of astrocyte precursors and reactive astrocytes (yellow cells) leading to gliosis could not be rescued and might lead to ON degeneration at adult stages. The majority of these defects are already present in *HET* but further worsened in *KO* mice, suggesting a gene dosage effect during development of the visual system (red gradient).

P28 is a consequence of astrocyte over-proliferation occurring at early prenatal stages, which might progressively impact on ON physiology and, ultimately, lead to neurodegeneration, as observed in other optic neuropathies (Carelli *et al*, 2017). This is also supported by the fact that ON demyelination but not gliosis can be rescued by Miconazole treatment in early postnatal life, suggesting that the two processes develop separately in this syndrome.

Our data disclose that gradual gliosis coupled to hypomyelination most likely contribute to the optic atrophy phenotype characterized in mutant mice and described in BBSOA patients. However, we cannot exclude that this atrophy might be caused by the altered ratio between proliferation and differentiation of RGCs during early

development, leading to a reduced number of fibers exiting the retina at the proper developmental time. We believe that the reduced number of differentiating RGCs upon *Nr2f1* downregulation represents only a delay in differentiation, which is gradually rescued. Indeed, the ON size between *WT*, *HET*, and *KO* animals at birth is only slightly affected, even in *KO* animals, but worsen at later stages (20% of reduction found in *HET* animals remains constant over time). We propose that this postnatal atrophy is due to transient RGC apoptosis at perinatal stages, combined with further axonal damage due to reactive gliosis. Indeed, Miconazole treatment recovers the oligodendrocyte number and myelination defects in *HET* ONs, but does not seem to rescue the overall size of

the ON, suggesting that atrophy is independent of the ON myelination grade. Thus, Nr2f1 is a strong regulator of astrocyte production and reduced dosage of *Nr2f1* results in astrocyte overproduction and gliosis ultimately affecting ON growth and stability.

Fourth, we found abnormal axonal conductance and impaired visual learning upon reduced *Nr2f1* dosage. OD malformation, optic nerve atrophy, and defective myelination of axonal fibers might ultimately lead to decreased signal transmission in our BBSOA model. In fact, by directly measuring the conduction velocity along the optic nerve and in following relay points along the visual system, we could demonstrate that *Nr2f1* deficiency affects normal visual conductance. Notably, in comparison with *WT* mice, *Nr2f1 HET* animals presented more than 20% delay in the latency of field potentials evoked in the dLGN, suggesting a deficit in the transmission velocity of retino-geniculate fibers. The deficit in axonal conduction velocity seems to be restricted to the ON, since the latency to field potentials evoked at the superior colliculus (1.65 ms; 10.6%) and to the visual cortex (2.1 ms; 12.6%) presented similar (not significantly different) delays.

BBSOA patients often display visual acuity and image processing deficits, besides ON atrophy (Bosch *et al*, 2014, 2016; Chen *et al*, 2016). Even if we focused here on the "periphery" of the visual system (the retina and ON), our data on the role of Nr2f1 in structures along the visual pathway, such as the thalamic dLGN and the visual cortex, unveil further abnormalities, possibly contributing to the visual deficits of *HET* mice and suggesting a causative link with patients' CVI. According to the present results, the performance of *Nr2f1 HET* mice in the Skinner box was mostly based on somatosensory and olfactory cues, since they failed to acquire the visual-dependent operant conditioning task. Nevertheless, the negative slope of their performance in the Skinner box could evince that this learning deficit is ascribed to more general limitations in the acquisition of complex learning tasks and not (exclusively) to visual difficulties. However, since *Nr2f1*-conditional mice have been proved to be capable of learning complex behavioral tasks (Flore *et al*, 2017), we preferentially advocate for a specific deficit in the perception and elaboration of visual stimuli at neocortical levels rather than a general defect in the learning process. Further studies will be necessary to specifically challenge the role of Nr2f1 in the development and functioning of thalamic and neocortical visual structures.

# Materials and Methods

### Animal procedures

All mouse experiments were conducted in accordance with relevant national and international guidelines and regulations (European Union rules; 2010/63/UE) and have been approved by the local ethical committee in France (CIEPAL NCE/2019-548) and Spain (JA/CAPD 06/03/2018/025). *Nr2f1* heterozygous (*HET*) and homozygous (*KO*) mice were generated and genotyped as previously described (Armentano *et al*, 2006). Littermates of *HET* and *KO* mice with normal *Nr2f1* alleles were used as control mice (herein called *WT*). Midday of the day of the vaginal plug was considered as embryonic day 0.5 (E0.5). Control and mutant mice were bred in a 129S2/SvPas background. Both male and female embryos and pups were used in this study; age is specified for each embryo/pup used

in specific experiments. Standard housing conditions were approved by local ethical committee in France (CIEPAL NCE/2019-548) and Spain (JA/CAPD 06/03/2018/025); briefly, adult mice were kept on a 12 h light–dark cycle and housed three per cage with the recommended environmental enrichment (wooden cubes, cotton pad, igloo) with food and water *ad libidum*.

### Immunofluorescence

Mouse embryonic heads were dissected and fixed in 4% paraformaldehyde (PFA) at 4°C for 3 h in agitation, then washed in PBS 1X and dehydrated in 25% sucrose overnight at 4°C. P8 brains and eyes were fixed by intra-cardiac perfusion of 4% PFA, then processed as previously described (Armentano *et al*, 2006, 2007). Primary antibodies used: NR2F1 (Abcam ab181137, 1:1,000, rabbit; R&D H8132, 1:1,000, mouse), ATPIF1 (Thermo Fisher 5E2D7, 1:500, mouse), Phospho-ERK (Cell signaling #4370, 1:200, rabbit), Sox10 (Santacruz sc-365692, 1:200, mouse), Sox2 (R&D AB2018, 1:500, mouse), NF1a (Abcam ab41851, 1:1,000, rabbit), MBP (Abcam ab7349, 1:1,000, rat), Brn3a (1:1,000, mouse, kind gift of Thomas Lamonerie, IBV, Nice), Cleaved Caspase3 (Cell signaling #9661, 1:2,000, rabbit), Pax2 (Abcam ab79389, 1:1,000, rabbit), Pax6 (Millipore AB2237, 1:500, rabbit), Ki67 (Thermo Fisher PA5-16446, 1:1,000), GLAST (1:1,000, rabbit), GFAP (Dako, 1:200, rabbit), S100β (Dako Cytomation Z0311, 1:200, rabbit), and Tuj1 (β-III TUBULIN, Covance MRB-435P, 1:1,000; or Sigma T8660, 1:1,000, mouse). Alexa Fluor 488, 555, 594, and 647 anti-mouse, anti-rabbit, or anti-rat IgG conjugates (Thermo Fisher scientific, all 1:500) were used as secondary antibodies.

### Nissl staining

Cryostat sections were left to dry for 60 min at room temperature. The slides were briefly washed in MilliQ water, post-fixed in 4% PFA 10 min at room temperature, and then immersed 5 min in the following staining solution: 0.025% thionine acetate (Sigma, T2029-5G), 0.025% Cresyl violet acetate (Sigma, T2029-5G), 100 mM sodium acetate, 8 mM acetic acid. Slides were then washed in ethanol/acetic acid (80/0.05%) for a variable time depending on the rate of clarification. Slides were dehydrated by two brief washes in ethanol 100% and finally 2 × 5-min washes in xylene. Sections were cover-slipped using Eukitt mounting medium (KO Kinder GmbH D-79110).

### *In situ* hybridization (ISH)

*Lmo4* RNA probes were *in vitro* transcribed from previously used plasmids (Alfano *et al*, 2014). Heads for ISH were fixed overnight with PFA 4% at 4°C, dehydrated in 25% sucrose, and embedded in OCT resin, then stored at −80°C. ISH was carried out on 14-μm cryosections, as previously described (Armentano *et al*, 2007).

### DiI labeling of mouse optic nerves

E18.5 mouse heads were fixed O/N in 4% PFA at 4°C. Lenses were gently removed from the eyes using sharp forceps, and a DiI crystal (Invitrogen, Life Technologies) was inserted inside the optic nerve head. To help keeping the crystal in position, the lens was placed

back inside the eye. Heads were kept in 4% PFA at 37°C for 3–4 weeks, allowing for DiI diffusion along the nerve. When ready to be processed, heads were washed in PBS, dehydrated in 25% sucrose, included in OCT, and 14-μm-thin sections were obtained by cryostat cutting. DAPI counterstaining was performed before slide mounting.

### Transmission electron microscopy

Eyes, with their ON attached, from P8 and P28 *WT* and *Nr2f1* mutants were fixed O/N at 4°C with 3% glutaraldehyde in PBS and then washed in PBS. ON were separated from the eye bulb and the surrounding muscle and treated with 1% osmium tetroxide in double-distilled water and 1% potassium ferrocyanide for 1 h at 4°C. After washing, the ONs were incubated in 0.15% tannic acid in 0.1 m PBS, pH 7.4 followed by staining with 2% uranyl acetate in distilled water for 1-h RT in darkness. After washing, the tissue was dehydrated with an ascending series of EtOH dilutions up to 100% at 4°C, followed by a stepwise incubation in propylene oxide. Tissue was then infiltrated stepwise with Epoxy resin (TAAB 812; TAAB Laboratories) and encapsulated in flat molds taking into accounts the proximo-distal orientation of the nerves. Blocks were polymerized for 48 h at 60°C. Semithin (1 μm) sections were stained with Toluidine blue, and ultrathin sections (70–80 nm) were collected along three different proximo-distal levels of the ONs, using an ultramicrotome (Leica Ultracut UCT) with a diamond blade (Diatome). Sections were collected on Cu-Pd boutonniere grids covered by Formvar and stained in drops of 2% aqueous uranyl acetate for 7 min, followed by Reynold's lead citrate for 2 min. Grids were analyzed and photographed with a JEM 1010 transmission electron microscope operated at 120 kV and recorded at different magnifications using a CMOS 4 K × 4 K, F416 of TVIPS camera. Images were taken to cover the entire major and minor axis of the nerve in order to have an equivalent representation of ON.

### Miconazole treatment

Miconazole (Sigma, QB-6813-5G) was resuspended in DMSO to a concentration of 200 μg/μl, aliquoted, and stored at −20°C. Mouse pups were treated with Miconazole (intra-peritoneal injection; 10 μg/g) between postnatal day (P) 2 and P8. The day of the injection, Miconazole stock was diluted 1,000× in physiological solution, to obtain a 0.2 μg/μl injectable solution.

### Collection and processing of human fetuses

All experiments involving the use of human samples conformed to the principles set out in the WMA Declaration of Helsinki and the Department of Health and Human Services Belmont Report. Tissues were made available in accordance with French bylaws (good practice concerning the conservation, transformation, and transportation of human tissue to be used therapeutically, published on December 29, 1998). Furthermore, the studies on human fetal tissue were approved by the French agency for biomedical research (Agence de la Biomédecine, Saint-Denis la Plaine, France, protocol n°: PFS16–002). Non-pathological human fetuses (11 and 14 gestational weeks, *n* = 2) were obtained from voluntarily terminated pregnancies after obtaining written informed consent from the parents (Gynaecology

Department, Jeanne de Flandre Hospital, Lille, France). Fetuses were fixed by immersion in 4% PFA at 4°C for 7 days. The tissues were then cryoprotected in 30% sucrose/PBS for 3 days, embedded in Tissue-Tek OCT compound (Sakura Finetek, USA), frozen in dry ice, and stored at −80°C until sectioning. Frozen samples were cut serially at 20 μm using a Leica CM 3050S cryostat (Leica Biosystems Nussloch GmbH, Germany).

### Electrophysiological recordings in behaving animals

Following the associative learning task, the same mice (13 *WT* and 18 *HET*) were prepared for the chronic recording of field potentials evoked at the visual pathway by flashes of light. To this end, animals were anesthetized with 4% chloral hydrate and stereotaxically implanted with two recording electrodes in the left lateral geniculate nucleus (2.06 mm posterior to Bregma, 2.0 lateral to the midline, and 2.5 mm depth from brain surface), the right superior colliculus (3.8 mm posterior to Bregma, 1.25 lateral to the midline, and 1.5 mm depth from brain surface), and the left visual cortex (3.8 mm posterior to Bregma, 2.5 lateral to the midline, and 0.5 mm depth from brain surface). Electrodes were made from 50-μm Teflon-coated tungsten wire (Advent Research Materials, Eynsham, UK). Two bare silver wires were affixed to the skull as ground. Electrodes were connected to two 4-pin sockets (RS-Amidata, Madrid, Spain) that were fixed to the cranium with dental cement. After surgery, animals were allowed 5 days for proper recovery.

Flashes of light were provided by a xenon arc lamp located 30 cm in front of the animal's eyes, and lasted ≈ 1 ms (Photic stimulator, Cibertec, Madrid, Spain). Photic stimulations were triggered from a programmable CS-20 stimulator (Cibertec). For recordings, each alert-behaving mouse was placed in a transparent box (5 × 5 × 5 cm), dark adapted for 30 min, and presented with a total of 40 stimuli at a rate of 6 per min. The recording box was located in the center of a larger (30 × 30 × 30 cm) cage made of polished aluminum and connected to a ground system to avoid unwanted electric interferences. Each animal received two stimulation sessions (Sanz-Rodriguez *et al*, 2018).

### Histology after electrophysiology

At the end of the electrophysiological recording experiments, mice were deeply re-anesthetized and perfused transcardially with saline and 4% PFA. Selected sections (50 μm) including the implanted electrodes were mounted on gelatinized glass slides and stained using the Nissl technique with 0.1% Toluidine blue, to determine the proper location of stimulating and recording electrodes. Photomicrographs were taken using a Leica DMRE microscope equipped with a Leica DFC550 camera and with the LAS V4.2 software (Leica Microsystems GmbH, Wetzlar, Germany). Photomicrograph reconstructions were made with the Microsoft Office Professional Plus 2010 and CorelDRAW X4 software.

### Behavior—instrumental conditioning

Operant conditioning took place in commercial Skinner box modules (*n* = 5) measuring 12.5 × 13.5 × 18.5 cm (MED Associates, St. Albans, VT, USA). The conditioning boxes were housed within sound-attenuating chambers (90 × 55 × 60 cm), which were

constantly illuminated (19 W lamp) and exposed to a 45-dB white noise (Cibertec, S.A., Madrid, Spain). Each Skinner box was equipped with a food dispenser from which pellets (MLabRodent Tablet, 20 mg; Test Diet, Richmond, IN, USA) can be delivered by pressing a lever. Each box was also equipped with a small light bulb located over the lever (Fig 6E). Before training, mice were handled daily for 7 days and food-deprived to 90% of their free-feeding weight (Madronal *et al*, 2010; Jurado-Parras *et al*, 2012). In all cases, training sessions lasted for 20 min. The start and end of each session was indicated by a tone (2 kHz, 200 ms, 70 dB) provided by a loudspeaker located in the recording chamber (Fig 6E).

In a first experimental step, mice (13 *WT* and 18 *HET*) were trained to press the lever to receive pellets from the food tray using a fixed-ratio (1:1) schedule (Fig 6E, top panel). Animals were maintained on the 1:1 schedule for 10 training sessions. Mice typically reach criterion (pressing the lever $\geq 20$ for two successive sessions) after 4–9 days of training (Madronal *et al*, 2010; Jurado-Parras *et al*, 2012), but for the sake of homogeneity, all mice where trained for 10 days.

After this preliminary training, mice were further conditioned using a small bulb light on/light off protocol for 10 additional days. In this program, only lever presses performed during the light period (20 s) were reinforced with a pellet (Fig 6E, bottom panel). The cued light was provided by the small bulb located over the lever. Lever presses performed during the dark period (20 ± 10 s) were not reinforced and restarted the dark protocol for an additional random (1–10 s) time. The light on/light off coefficient was calculated as follows: (number of lever presses during the light period − number of lever presses during the dark period)/total number of lever presses). In this case, the criterion was to reach a positive performance (i.e. a larger number of lever presses during the light on than during the light off periods) for two successive sessions. Conditioning programs, lever presses, and delivered reinforcements were controlled and recorded by a computer, using a MED-PC program (MED Associates, St. Albans, VT, USA).

### Statistical analysis

All the data were statistically analyzed and graphically represented using Microsoft Office Excel software. Quantitative data are shown as the mean ± standard error (SEM). For operant conditioning, statistical significance of differences between groups was inferred by two-way ANOVA (sessions by groups) for repeated measures (sessions), with a contrast analysis (Holm-Sidak method) for a further study of significant differences. The best fit for a linear equation was settle for each group across the operant conditioning sessions. Data collected from the electrophysiological experiments were subjected to normality (Shapiro–Wilk) and equal variance tests and then compared using a Student's *t*-test between control and experimental groups. The same test and procedure were used to check differences on the percentage of lever responses between the light and dark periods in both mouse groups.

For cell percentage/number quantification after immunofluorescence (IF), measurements were performed on at least nine sections coming from 3 to 5 different animals, unless otherwise stated. To minimize subjective bias, sample identity (e.g. genotypes) was randomized by associating an identification number to each sample before processing. Fixed embryos with damaged tissues were excluded from any further analysis/processing. Microscope images were processed with Photoshop or ImageJ software, by randomly overlapping fixed-width (100 μm) rectangular boxes on the area of interest (e.g. the retina), then quantifying positive cells inside the boxes. For ON cross-sections, the whole ON area was considered for the analysis, without placing any box. In the specific case of EM images, the myelination grade was evaluated as the average number of myelinated fibers found inside randomly placed 100-μm$^2$ boxes. When calculating percentages over the total cell number, the latter was quantified by counting DAPI$^+$ nuclei, unless otherwise specified. Data were compared by two-tailed Student's *t*-test (when comparing two data group) or by ANOVA (analysis of variance; for comparison of three or more groups) and statistical significance was set as follows: *$P \leq 0.05$; **$P \leq 0.01$; ***$P \leq 0.001$. While Figure legends only show *P*-value ranges, exact *P*-values can be found in the corresponding Appendix Table S1 and/or in the main text.

**Expanded View** for this article is available online.

### Acknowledgements

We are grateful to Eya Setti for technical help and to the whole Studer laboratory for feedback on the study. We thank Simona Casarosa and Federico Cremisi for useful hints during the initial design of the research study. We thank Ms. M. Sanchez-Enciso and Mr. J.M. González-Martín for their help in animal handling and care, as well as for instrumental design for electrophysiology and behavioral experiments. We also thank the iBV PRISM Microscopy facility for their regular support, and the iBV animal facility, particularly Kevin Moneret, for animal care. The help of Milagros Guerra and M. Teresa Rejas of the CBMSO EM facility is also acknowledged. This work was supported by an ERA-NET Neuron II grant (Improv-Vision) ANR-15-NEUR-0002-04 and by "Investments for the Future" LabEx SIGNALIFE (grant ANR-11-LABX-0028-01) to M.S., and by a postdoctoral fellowship from the Ville de Nice, France ("Aides Individuelles aux Jeunes Chercheurs") to M.B. Spanish Ministry of Science and Innovation BFU2016-75412-R (with FEDER funds), PCIN-2015-176-C02-01/ERA-Net NeuronII and an Institutional CBMSO Grant from the Fundación Ramón Areces to P.B. The electrophysiological and behavioral study was supported by grants from the Spanish Ministry of Science and Innovation (BFU2017-82375-R), the Junta de Andalucía (Spain, BIO-122), and the Spanish Tatiana Pérez de Guzmán el Bueno Foundation to A.G. and J.M.D.-G.

### Author contributions

MS and MB conceptualized and designed the research study. AG and JMD-G designed, conducted, and analyzed the electrophysiological/behavioral experiments. MB, PK, CA, and LS conducted the experiments and acquired the data. PG provided the human samples. MB, MS, and PB analyzed the data. MB, MS, and PB wrote the paper.

### Conflict of interest

The authors declare that they have no conflict of interest.

### For more information

BBS optic atrophy (BBSOA) syndrome resulting from *NR2F1* deletions/mutations is now listed as a rare genetic disease (*OMIM: #615722; ORPHANET: #401777*). Updated information concerning BBSOA patients and their families can be found on the NR2F1 Foundation website (https://www.nr2f1.org; or: https://www.facebook.com/NR2F1/).

## The paper explained

### Problem

Optic nerve (ON) atrophy denotes the loss of part or all the retinal ganglion cell (RGC) fibers in the optic nerve, leading to reduced visual acuity and gradual vision loss. As recently shown, mutations/deletions in the *NR2F1* locus lead to a rare hereditary form of ON atrophy associated with intellectual disability, epilepsy, and autism, named Bosch-Boonstra-Schaaf optic atrophy (BBSOA) syndrome. Despite the increasing number of patients identified in the last few years, the molecular and cellular etiology of BBSOA syndrome remains elusive, due to the absence of a suitable animal model.

### Results

In this study, we report that *Nr2f1* mutant mice (both *heterozygotes —HET* and *homozygotes—KO*) recapitulate key features of BBSOA patients, such as optic disc malformations, optic nerve atrophy, and cerebral visual impairment. By taking advantage of the mouse model, we identified multiple cellular features, such as delayed RGC differentiation and apoptosis, decreased ON myelination, and increased astrogliosis, ultimately resulting in reduced axonal conduction velocity from the retina to higher order centers. At adult stages, *Nr2f1 HET* mice have visual and associative learning deficits, reminiscent of the cerebral visual impairment described in patients.

### Impact

Our findings show that *Nr2f1* mutant mice can be used as a model to reproduce the BBSOA syndrome and, more broadly, can serve as a tool to test possible therapeutic approaches aimed at counteracting ON neuropathies. As a proof of concept, we tested Miconazole as a chemical drug to rescue demyelination defects in early postnatal pups. Miconazole restores appropriate oligodendrocyte number and myelination levels. If proved efficient and safe in humans, Miconazole could be used in the future as a potential treatment to improve ON signal conductance in BBSOA patients.

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
