## [Review Process File · EMBO Molecular Medicine]

Mouse *Nr2f1* haploinsufficiency unveils new pathological mechanisms of a human optic atrophy syndrome

Michele Bertacchi, Agnès Gruart, Polynikis Kaimakis, Cécile Allet, Linda Serra, Paolo Giacobini, José M. Delgado-García, Paola Bovolenta and Michèle Studer.

Review timeline:

Submission date:	8 th January 2019
Editorial Decision:	15 th February 2019
Revision received:	30 th April 2019
Editorial Decision:	23 rd May 2019
Revision received:	5 th June 2019
Accept:	12 th June 2019

Editor: Jingyi Hou

Transaction Report:

1st Editorial Decision

11th February 2019

Thank you for the submission of your manuscript to EMBO Molecular Medicine. We have now heard back from the three referees whom we asked to evaluate your manuscript.

You will see from the comments below that the referees find the manuscript to be of interest and are supportive of publication. While referee #2 is more enthusiastic about the findings, Referee #1 requests additional controls and experiments to strengthen the findings and make them more conclusive. We would expect a point-by-point response to all concerns raised by the two referees, along with providing additional details and clarifications. Importantly, we would like you to discuss the brain structure change in NR2F1 +/- mice as commented by referee #1. We also expect you to provide mitochondrial function/dysfunction data as referee #2 suggested.

We would welcome the submission of a revised version within three months for further consideration and would like to encourage you to address all the criticisms raised as suggested to improve conclusiveness and clarity.

REFeree REPORTS

Referee #1 (Remarks for Author):

In general, the Bosch-Boonstra-Schaaf optic atrophy (BBSOA) is caused by loss of function of the NR2F1 gene. Using a knockout mouse model that is heterozygous for the NR2F1 gene, this paper describes a phenotype in the mouse model that is comparable to that of the human disease. Moreover, histological analysis of the retina, the brain in different stages of development in conjunction with an interventional assay an insight into the pathomechanisms was possible. Although this description of the mouse model is of importance to develop a possible therapy there are some of the conclusions not fully supported by the data.

Major comments

1. Effect of Miconazole: Recent studies showed that miconazole has the potential to reverse neuronal degeneration by increasing myelination by oligodendrocytes. It seems that this effect is based on ERK signaling. The authors found comparable results in their model albeit miconazole improves the oligodendrocyte phenotype but not the astrocytosis. The authors conclude that NR2F1 might act in several independent pathways. However, the data do not show that miconazole is reversing a NR2F1-dependent effect in oligodendrocytes. The authors just describe a phenomenon that can result from interference with secondary effects and not by NR2F1-specific changes in the oligodendrocytes. Thus, NR2F1 specific conclusions are not possible without showing more specific data (e.g. changes in gene expression, signaling by the ERK pathway).
2. Retinoic acid: NR2F1 interferes with retinoic acid growth arrests. The interference with retinoic acid signaling might be one of the likely explanations for the changes in the ganglion cell axons because this signaling is required for ganglion cell axon guidance during embryonic development.
3. Analysis of light-evoked neuronal activity along the ganglion cell to visual cortex axis. The test is a very rough test and might be technically biased by the recording. First, the authors need to show that the Ganzfeld-ERG is normal in these animals especially the b-wave kinetics (is here already a shift in the latency?). Single flash data recorded in the visual cortex can have very variable waveforms. Better would have been to use checkerboard stimulation that directly leads to ganglion cells-driven recordings. The effects shown here are moderate. A control from knockout mice would be helpful. Further, the waveform amplitudes are not very different. Thus, it cannot be concluded that NR2F1 haplo-insufficiency "hampers" visual processing because the latency is only one sign.
4. Skinner box studies: The last part of the conclusion that the skinner box data might point to light-dependent effect in NR2F1^{+/-} mice by the optic nerve properties in these animals is not supported at all. The changes in the brain are too severe to relate the data only to malfunction of the optic nerve because learning needs the full brain. A recent paper even suggested hearing impairment.

Referee #2 (Remarks for Author):

This is a very well written paper detailing the elegant findings in a mouse model of NR2F1 and showing a very cogent set of data that suggest a very important role in development and differentiation as well as myelination. The data has a very high technical standard and is presented very clearly with very detailed figures and strong data. This is highly novel and of great interest to the field. Although the condition in humans may be rare it is very impactful to gain a better insight into the pathophysiology as there are some interesting new mechanisms explored. The model system is as fit for purpose as ever mice can be.

I would be very keen to see some data on mitochondrial function/ dysfunction - as there is data referenced (ref 8) but no comment is made on this and no experiments here have looked at the possibility that mitochondrial dysfunction may be important to or contribute to the phenotype. This should at the very least be discussed as a contributory aspect of the pathophysiology or some data presented to show whether the mouse does indeed reflect any mitochondrial dysfunction.

See next page.

RESPONSES TO REFEREES POINT BY POINT

Referee#1:

1. Effect of Miconazole: Recent studies showed that miconazole has the potential to reverse neuronal degeneration by increasing myelination by oligodendrocytes. It seems that this effect is based on ERK signaling. The authors found comparable results in their model albeit miconazole improves the oligodendrocyte phenotype but not the astrocytosis. The authors conclude that NR2F1 might act in several independent pathways. However, the data do not show that miconazole is reversing a NR2F1-dependent effect in oligodendrocytes. The authors just describe a phenomenon that can result from interference with secondary effects and not by NR2F1-specific changes in the oligodendrocytes. Thus, NR2F1 specific conclusions are not possible without showing more specific data (e.g. changes in gene expression, signaling by the ERK pathway).

Response:

We thank the reviewer for the insightful remark. To respond to the reviewer's query, we have immunostained Miconazole-treated *Nr2f1* heterozygous optic nerves with p-ERK and quantified expression levels in the optic nerve and in Sox10+ (oligodendrocyte marker) cells. As expected, levels of p-ERK expression are significantly increased in Miconazole-treated mutants compared to non-treated HETs or WT. We have now added this result in Fig. 5 (F-I).

2. Retinoic acid: NR2F1 interferes with retinoic acid growth arrests. The interference with retinoic acid signaling might be one of the likely explanations for the changes in the ganglion cell axons because this signaling is required for ganglion cell axon guidance during embryonic development.

Response:

Thank you for this interesting comment. Retinoic acid (RA) induces mainly differentiation of neural cells by increasing *Nr2f1* expression, and overexpression of *Nr2f1* seems to interfere with growth arrest, as previously shown in ES cell lines (Zhuang & Gudas, 2008). But since we are working with *null* and *heterozygotes* in which *Nr2f1* levels are reduced, and since no clear evidence show that *Nr2f1* acts directly on RA signaling, we do not really foresee any RA interference in the optic nerve phenotype of *Nr2f1* mutant eyes. Moreover, from what we understand, exposure of RA in combination with BDNF increases the number, but not the length of neurites in chick retinal ganglion explants (Mey & Rombach, 1999), differently from embryonic DRGs, sympathetic, spinal cord and olfactory neurons in which RA seems to be involved in axonal elongation (reviewed in (Clagett-Dame et al, 2006). However, RA seems more suited to guide developing neurites towards regions of higher RA concentration, thus acting as a directional cue. It is unlikely that the optic atrophy phenotype, we describe in this study, is due to altered RA signaling during RGC axonal growth and/or guidance, also because *Nr2f1*-deficient RGC axons do reach the dLGN in the thalamus, although in reduced number. Nevertheless, we agree with the reviewer that a careful investigation of *Nr2f1* action on RA signaling might be worth pursuing in a future study.

3. Analysis of light-evoked neuronal activity along the ganglion cell to visual cortex axis. The test is a very rough test and might be technically biased by the recording. First, the authors need to show that the Ganzfeld-ERG is normal in these animals especially the b-wave kinetics (is here already a shift in the latency?). Single flash data recorded in the visual cortex can have very variable waveforms. Better would have been to use checkerboard stimulation that directly leads to ganglion cells-driven recordings. The effects shown here are moderate. A control from knockout mice would be helpful. Further,

the waveform amplitudes are not very different. Thus, it cannot be concluded that NR2F1 haplo-insufficiency "hampers" visual processing because the latency is only one sign.

Response:

We appreciate the reviewer's concern, but we would like to point out that our main goal was to determine putative deficits in the latency of evoked field potentials at selected recording sites across the visual pathway, in accordance with the demyelination optic nerve phenotype. We fully agree with the reviewer that a more detailed analysis of evoked field potentials would need more sophisticated instruments. Nevertheless, we believe that our approach is accepted by the scientific community, since we consistently used this visual stimulation system in behaving cats (Gruart et al., *Brain*, 2003), rats (Barriga-Rivera et al., *Conf Proc IEEE Eng Med Biol Soc.* 2018) and mice (Sanz-Rodriguez et al., *J. Neurosci.*, 2018).

Regarding the reviewer's point of using *knockout* mice as controls, which would certainly lead to a stronger phenotype, is unfortunately not possible, since *KO* mice die around P8. Overall, we believe that our recordings show that the decrease in conduction velocity is concentrated at the optic nerve (Fig. 6C), a finding supported by the molecular and cellular analysis presented in Figs. 1-5 of this study. Hence, we introduced the following comment on page 11: "*Since these differences were similar (≈ 2 ms) among the three recording sites we can assume that the deficits in axonal conduction velocity are mainly restricted to the optic nerve.*" We have also down-tone the statement that "NR2F1 haploinsufficiency hampers visual processing" by changing it with "affects normal visual conductance".

4. Skinner box studies: The last part of the conclusion that the skinner box data might point to light-dependent effect in NR2F1+/- mice by the optic nerve properties in these animals is not supported at all. The changes in the brain are too severe to relate the data only to malfunction of the optic nerve because learning needs the full brain. A recent paper even suggested hearing impairment.

Response:

To address the reviewer's concern, we have included more available data collected during the first instrumental conditioning test, and carried out in completely illuminated Skinner boxes. These new data (included in Fig. 6E-G) show that in this situation, *Nr2f1 HET* mice acquired this complex associative learning task at the same rate (Fig. 6F) and with a similar performance (Fig. 6G) than their WT littermates. Acquisition rates were instead significantly different when the mutant mice had to perform the task in the presence of a weak light bulb (Fig. 6E, H, I). Although other CNS deficits cannot be completely excluded, these experiments point to visual impairments. The text has been amended in the Methods, Results and Figure legend sections to introduce these new data.

Finally, to respond to possible hearing anomalies (described only in some human patients), we would like to clarify that the tone just indicated the beginning and the end of the experimental session, together with the activation/deactivation of the whole recording system (placing or removing the animal, switching general lights on/off, etc.). Thus, the outcome of this test did not depend on hearing perception.

Referee#2:

This is a very well written paper detailing the elegant findings in a mouse model of NR2F1 and showing a very cogent set of data that suggest a very important role in development

and differentiation as well as myelination. The data has a very high technical standard and is presented very clearly with very detailed figures and strong data. This is highly novel and of great interest to the field. Although the condition in humans may be rare it is very impactful to gain a better insight into the pathophysiology as there are some interesting new mechanisms explored. The model system is as fit for purpose as ever mice can be.

Response:

We thank this reviewer for the very positive and encouraging comments on our study.

I would be very keen to see some data on mitochondrial function/ dysfunction - as there is data referenced (ref 8) but no comment is made on this and no experiments here have looked at the possibility that mitochondrial dysfunction may be important to or contribute to the phenotype. This should at the very least be discussed as a contributory aspect of the pathophysiology or some data presented to show whether the mouse does indeed reflect any mitochondrial dysfunction.

Response:

We fully agree with the reviewer and thank him for asking some evidence of potential mitochondrial dysfunction in the optic nerve of *Nr2f1* mutant mice. We have now added new experimental data in Suppl. Fig. 5. In this figure, we show that while mitochondria have an abnormally larger morphology and affected distribution of cristae in reactive astrocytes of P8 mutants (Suppl. Fig. 5A-D), this does not seem to be the case for optic nerve fibers. However, we found high levels of ATP1F1, an ATPase mitochondrial inhibitor, in retinal ganglion cells of *Nr2f1* *HET* retinae, already at P8 (data not shown). In the new figure, we added the staining at adult stages since RGCs are more dispersed and ATP1F1 can be better visualized (Suppl. Fig. 5G-H'). Our hypothesis is that mitochondrial dysfunction might have induced a surface expansion and/or increased mitochondrial biogenesis. At this point, we do not know yet whether this phenotype might contribute to the optic nerve pathophysiology we observed in the *HET* animals or is just a secondary consequence, due for example to increased astrogliosis. We have now discussed this aspect in the text.

References:

Barriga-Ribera A, Suaning GJ, Delgado-García, JM, Gruart A (2018) Optic nerve and retinal electrostimulation in rats: direct activation of the retinal ganglion cells. *Proceedings 40th Conference of the IEEE-EMBC, 2018*, pp. 1226-1229. doi: 10.1109/EMBC.2018.8512517.

Clagett-Dame M, McNeill EM, Muley PD (2006) Role of all-trans retinoic acid in neurite outgrowth and axonal elongation. *Journal of neurobiology* **66**: 739-756

Gruart A, Streppel M, Guntinas-Lichius O, Angelov DN, Neiss WF, Delgado-García JM (2003) Motoneuron adaptability to new motor tasks following two types of facial-facial anastomosis in cats. *Brain* **126**:115-133.

Mey J, Rombach N (1999) Retinoic acid increases BDNF-dependent regeneration of chick retinal ganglion cells in vitro. *Neuroreport* **10**: 3573-3577

Sanz-Rodriguez M, Gruart A, Escudero-Ramirez J, de Castro F, Delgado-García JM, Wandosell F, Cubelos B (2018) R-Ras1 and R-Ras2 Are Essential for Oligodendrocyte Differentiation and Survival for Correct Myelination in the Central Nervous System. *Journal of Neuroscience* **38**: 5096-5110

Zhuang Y, Gudas LJ (2008) Overexpression of COUP-TF1 in murine embryonic stem cells reduces retinoic acid-associated growth arrest and increases extraembryonic endoderm gene expression. *Differentiation* **76**: 760-771

Thank you for the submission of your revised manuscript to EMBO Molecular Medicine. We have now received the enclosed report from the referee who was asked to re-assess it. As you will see the reviewer is now overall supportive and I am pleased to inform you that we will be able to accept your manuscript pending the following final amendments:

REFeree REPORTS.

Referee #1 (Comments on Novelty/Model System for Author):

A mouse model with high relevance for a hereditary nervus opticus atrophy has been thoroughly investigated by state of the art Methods. Referee's concerns were adequately answered and the manuscript has improved. The interventional data Show prove of principle for drug Treatment, however, the possible Translation Needs to be proven.

Referee #1 (Remarks for Author):

The authors have carefully revised and improved their manuscript.